# Thermal proteome profiling of breast cancer cells reveals proteasomal activation by CDK4/6 inhibitor palbociclib

Teemu P Miettinen[1,2,3,4,‡], Julien Peltier[2,5,‡], Anetta Härtlova[2,5], Marek Gierliński[6], Valerie M Jansen[7,†], Matthias Trost[2,5,*] (ID) & Mikael Björklund[1,**]

## Abstract

Palbociclib is a CDK4/6 inhibitor approved for metastatic estrogen receptor-positive breast cancer. In addition to G1 cell cycle arrest, palbociclib treatment results in cell senescence, a phenotype that is not readily explained by CDK4/6 inhibition. In order to identify a molecular mechanism responsible for palbociclib-induced senescence, we performed thermal proteome profiling of MCF7 breast cancer cells. In addition to affecting known CDK4/6 targets, palbociclib induces a thermal stabilization of the 20S proteasome, despite not directly binding to it. We further show that palbociclib treatment increases proteasome activity independently of the ubiquitin pathway. This leads to cellular senescence, which can be counteracted by proteasome inhibitors. Palbociclib-induced proteasome activation and senescence is mediated by reduced proteasomal association of ECM29. Loss of ECM29 activates the proteasome, blocks cell proliferation, and induces a senescence-like phenotype. Finally, we find that ECM29 mRNA levels are predictive of relapse-free survival in breast cancer patients treated with endocrine therapy. In conclusion, thermal proteome profiling identifies the proteasome and ECM29 protein as mediators of palbociclib activity in breast cancer cells.

**Keywords** breast cancer; CDK4; palbociclib; proteasome; thermal proteome profiling

**Subject Categories** Cancer; Molecular Biology of Disease; Post-translational Modifications, Proteolysis & Proteomics

**The EMBO Journal (2018) 37: e98359**

See also: **RL de Oliveira & R Bernards** (May 2018)

## Introduction

The two closely related cyclin-dependent kinases CDK4 and CDK6 together with their binding partners cyclin D1, D2, and D3 are key regulators of growth and cell proliferation. The main mechanism by which CDK4/6–cyclin D complexes drive cell cycle progression is through phosphorylation of the retinoblastoma protein (RB1) and related p107 and p130 proteins, which are among the few known CDK4/6 substrates (Anders *et al*, 2011). The phosphorylation of RB1 drives cell cycle progression from G1 to S phase by allowing E2F transcription factor to induce coordinated transcription of genes required for DNA synthesis and cell cycle progression. Nonetheless, cyclin D and CDK4/6 are not universally required for cell cycle progression in most cell types during development (Kozar & Sicinski, 2005).

Breast cancer is the most common cancer in women with close to two million cases per year globally. CDK4/6 inhibition has been recognized to have therapeutic potential in estrogen receptor-positive (ER+) breast cancers, which constitute up to 75% of all breast cancers. In breast tissue, the CDK4/cyclin D1 axis appears critical for cancer initiation, as demonstrated using animal models and analysis of human breast cancer specimens (Bartkova *et al*, 1994; Landis *et al*, 2006; Yu *et al*, 2006; Choi *et al*, 2012). In addition, CDK4/cyclin D inhibition may be additionally beneficial for the treatment of human epidermal growth factor receptor 2 (HER2) and RAS-driven breast cancers (Yu *et al*, 2001; Goel *et al*, 2016) and potentially many other cancers because cyclin D1 displays most frequent somatic copy number aberrations of all cancer genes (Leiserson *et al*, 2015).

Palbociclib (PD-0332991/Ibrance) is a potent and selective CDK4/6 inhibitor based on analysis of 36 other kinases tested (Fry *et al*, 2004; Toogood *et al*, 2005). In cell models, palbociclib

1 Division of Cell and Developmental Biology, University of Dundee, Dundee, UK
2 MRC Protein Phosphorylation and Ubiquitylation Unit, University of Dundee, Dundee, UK
3 MRC Laboratory for Molecular Cell Biology, University College London, London, UK
4 Koch Institute for Integrative Cancer Research, Massachusetts Institute of Technology, Cambridge, MA, USA
5 Institute for Cell and Molecular Biosciences, Newcastle University, Newcastle upon Tyne, UK
6 Division of Computational Biology, University of Dundee, Dundee, UK
7 Division of Hematology-Oncology, Vanderbilt University Medical Center, Nashville, TN, USA
*Corresponding author. Tel: +44 191 2087009; E-mail: matthias.trost@ncl.ac.uk
**Corresponding author. Tel: +44 1382 388469; E-mail: mikael.bjorklund.lab@gmail.com
‡ These authors contributed equally to this work
† Present address: Eli Lilly and Company, Indianapolis, IN, USA

displays preferential activity in ER$^+$ as well as HER2-amplified cell lines (Finn *et al*, 2009). In a recent phase 3 trial, the combinatory treatment with palbociclib and fulvestrant, a selective estrogen receptor degrader, more than doubled the median progression-free survival as compared with fulvestrant alone (Turner *et al*, 2015; Cristofanilli *et al*, 2016). While broader specificity CDK inhibitors have performed poorly in clinical trials, it is not entirely clear why a selective CDK4/6 inhibitor displays such a remarkable therapeutic response (Sherr *et al*, 2015). In addition to the blockade of cancer cell proliferation, palbociclib also induces senescence (Anders *et al*, 2011) and displays other unexpected responses such as reduction in tumor volume without inducing apoptosis (Choi *et al*, 2012). Many RB1-deficient breast cancer cells display reduced DNA replication in response to palbociclib (Dean *et al*, 2010), and reduced levels of phosphorylated RB1 alone do not explain the oncosuppressive effects of palbociclib, at least in liposarcoma patients where such data are available (Kovatcheva *et al*, 2015). Similarly, palbociclib-mediated growth inhibition cannot be fully recapitulated by knockdown of CDK4/6 in breast cancer cells (Vijayaraghavan *et al*, 2017). Thus, while the RB1 pathway is clearly important for CDK4/6-mediated cell cycle progression, the therapeutic response to palbociclib potentially includes additional CDK4/6- or RB1-independent activities. Consistently, some of the cellular responses to palbociclib, such as senescence, have been reversed or augmented by knockdowns of seemingly unrelated proteins (Jansen *et al*, 2017; Rapisarda *et al*, 2017), suggesting that the mode of action of palbociclib is more complex than previously appreciated. Palbociclib is therefore likely to have unanticipated, clinically relevant targets beyond cell cycle inhibition. Furthermore, clinical trials with palbociclib have evaluated the role of possible biomarkers, but, for example, gain of cyclin D1, loss of p16, PI-3 kinase mutations, or hormone-receptor expression level are not predictive of the treatment response (Finn *et al*, 2015; Cristofanilli *et al*, 2016). Thus, there is an urgent need to identify biomarkers that could predict the patient population who would benefit most from palbociclib.

Cellular thermal shift assay (CeTSA) has recently emerged as a novel method to identify drug engagement with target proteins in live cells (Martinez Molina *et al*, 2013; Miettinen & Bjorklund, 2014; Savitski *et al*, 2014; Huber *et al*, 2015; Reinhard *et al*, 2015). This method is based on observing the changes in the thermal stability of a protein, which may be due to direct drug binding, drug-induced conformational changes, binding to other cellular components or post-translational modifications such as phosphorylation. When combined with mass spectrometry, the thermal stability changes in proteins can be quantified using a quantitative proteomic approach, holding great promise for identifying drug targets in live cells (Savitski *et al*, 2014; Franken *et al*, 2015; Becher *et al*, 2016). Furthermore, alternative sample preparation with NP-40 detergent has allowed improved detection of membrane proteins (Reinhard *et al*, 2015).

Here, we applied mass spectrometry-based cellular thermal shift assay (MS-CeTSA) also known as thermal proteome profiling to study the effects of palbociclib in MCF7 breast cancer cells and identified the proteasome as a downstream target of palbociclib. Palbociclib reduced the association of proteasomal accessory protein ECM29 with the proteasome resulting in enhanced proteolysis. Consistent with the notion that proteasome activation is part of the mode of action for palbociclib, inhibition of proteasome suppressed the cell cycle and senescence phenotypes induced by palbociclib. Together, these results show that thermal proteome profiling can be used to discover novel and unexpected drug responses with potential implications to clinical research.

## Results

### Thermal proteome profiling in MCF7 cells

We performed thermal proteome profiling in MCF7 cells, an ER$^+$ and HER2$^-$ cell line with sensitivity to palbociclib (Finn *et al*, 2009). Palbociclib was used at 10 μM concentration in order to ensure a complete saturation of ligand binding, which maximizes the shift in detected protein denaturation temperature ($\Delta T_m$; Martinez Molina *et al*, 2013; Miettinen & Bjorklund, 2014; Savitski *et al*, 2014; Huber *et al*, 2015; Reinhard *et al*, 2015). While a 10 μM concentration of palbociclib is higher than the typical IC$_{50}$ for long-term growth inhibition in cell culture, such concentrations are present in tumors in preclinical mouse models (Nguyen *et al*, 2010). MCF7 cells were treated with palbociclib or vehicle (H$_2$O) in triplicates for 1 h and incubated at temperatures ranging from 37 to 65°C (Fig 1A). Heat-aggregated proteins were removed by centrifugation, and soluble proteins were analyzed by multiplexed quantitative mass spectrometry using isotopically labeled tandem mass tags as in Savitski *et al* (2014).

Typical data normalization approaches are not easily applicable for thermal proteome profiling data because the total amount of soluble protein declines with temperature. We previously described that the accuracy of Western blot-based cellular thermal shift assay is improved by normalization to thermally stable proteins, such as superoxide dismutase 1 (SOD1; Miettinen & Bjorklund, 2014). When equal amount of protein is analyzed by mass spectrometry for each temperature point, thermally stable proteins display increased abundance with temperature in the raw data (Fig 1A, middle). We used a set of 32 identified, abundant, and thermally stable proteins for normalization of the mass spectrometry data (Table EV1). This single-step normalization improved the data quality substantially, reducing the median standard error of the denaturation temperature for the measured proteome from 1.1 to 0.6°C (Fig EV1A).

Our analysis quantified 5,515 proteins with high confidence. For 3,707 of these, we could obtain high-quality thermal denaturation curves (Table EV2). The remaining proteins with poorer quality thermal denaturation curves were enriched for membrane proteins and mitochondrial proteins as shown before (Savitski *et al*, 2014) or displayed atypical thermal denaturation profiles (see Fig EV1B for examples). A global denaturation profile of the MCF7 proteome indicated a denaturation catastrophe at 49°C (Fig 1A, right), consistent with the cellular "thermal death point" of mesophiles (Ghosh & Dill, 2010) and the thermal stability of the erythroleukemia K562 proteome (Savitski *et al*, 2014; Fig EV2).

We next analyzed how palbociclib affects thermal stability of proteins, first focusing on 195 quantified protein kinases (Table EV3). Palbociclib increased thermal stability ($\Delta T_m$ describes

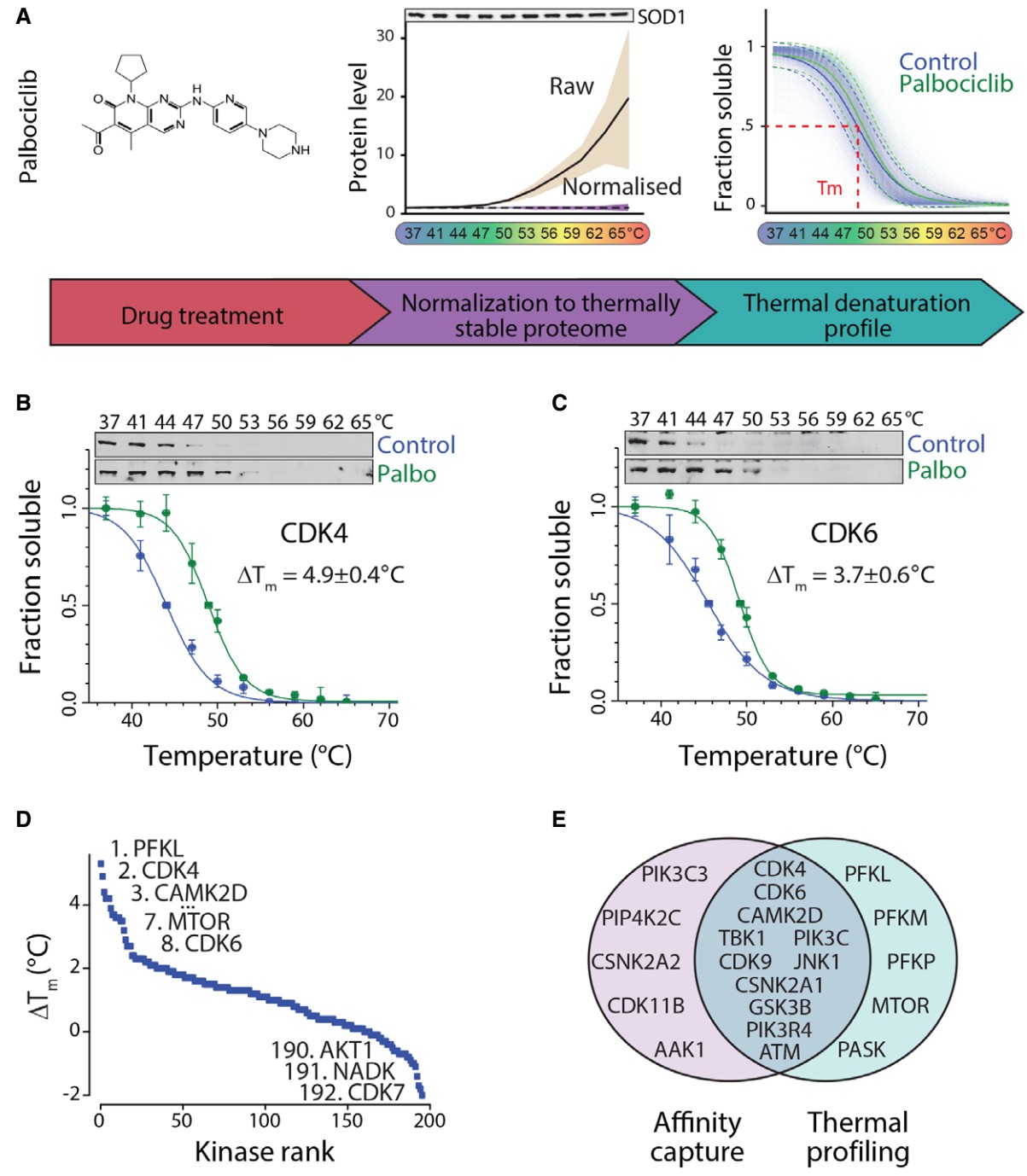

**Figure 1.  Thermal proteome profiling of MCF7 breast cancer cells treated with palbociclib.**

A      Schematic presentation of the experimental setup. Left, MCF7 cells were treated with 10 μM palbociclib for 1 h after which samples were incubated at temperatures between 37 and 65°C. Soluble proteins were analyzed from each fraction using quantitative mass spectrometry. Middle, Increased levels of 32 abundant thermally stable proteins in raw mass spectrometry data. This set is then used for normalization. The black line displays mean protein levels of the 32 proteins with beige shading displaying 95% confidence interval. The dashed line is the mean levels after normalization with purple shading displaying 95% confidence interval. Western blot displays the levels of one of the thermally stable proteins, SOD1. Right, Thermal denaturation curves obtained from all proteins in control (blue) and palbociclib-treated (green) samples. Dashed lines indicate the standard deviation (SD) of the global denaturation curve (solid line).

B, C    Thermal denaturation curves and Western blots of the main palbociclib targets CDK4 (B) and CDK6 (C) showing thermal stabilization upon palbociclib addition.

D      Palbociclib-induced thermal shifts ($\Delta T_m$) of all identified kinases.

E      Venn diagram displaying the overlap between kinases displaying palbociclib-induced positive $\Delta T_m$ and kinases identified as direct palbociclib targets by affinity purification (Sumi *et al*, 2015).

Data information: In panels (B and C), data are presented as means ± SEM from three individual biological replicates.
Source data are available online for this figure.

the difference between the palbociclib treatment and control melting temperatures) of its main targets CDK4 and CDK6, as expected (Fig 1B and C). While the magnitude of the $\Delta T_m$ does not directly indicate binding affinity (Martinez Molina *et al*, 2013), CDK4 displayed the second strongest $\Delta T_m$ of all kinases, exceeded only by phosphofructokinase (PFKL, liver isoform), a recently identified CDK6 substrate (Wang *et al*, 2017; Fig 1D). $\Delta T_m$ of CDK6 was the 8th strongest. The mechanistic target of rapamycin (mTOR) was also among the kinases with high $\Delta T_m$ value. We also observed thermal destabilization, a decrease in $\Delta T_m$, which could result from indirect drug effects (Franken *et al*, 2015), ligand binding to partially unfolded state (Cimmperman *et al*, 2008) or conformational changes that destabilize the protein structure. The most destabilized kinases by palbociclib included CDK7 and AKT1 (Figs 1D, and EV3A and B). Overall, we observed substantial overlap with previously identified palbociclib binding kinases (Sumi *et al*, 2015; Fig 1E) and previously measured palbociclib inhibitory activity against 22 kinases *in vitro* (Fig EV3C), indicating high quality of our dataset. While our data indicated that palbociclib may affect multiple pathways including the PI3K/AKT/mTOR signaling pathway (Fig EV3D), the PI3K/AKT/mTOR pathway inhibition was weak and evident only at higher drug concentrations (Fig EV3E–G). These findings are consistent with previous observations that CDK4/6 inhibition can partially attenuate mTORC1 activity (Goel *et al*, 2016).

## Palbociclib activates the 26S proteasome

To gain a global insight into palbociclib-induced cellular changes, we analyzed what protein complexes were affected by palbociclib. To visualize the separation from non-affected complexes, we plotted changes in both $\Delta T_m$ and $\Delta S$, a parameter related to the change in the slope of the melting curve (see Materials and Methods). This analysis identified complexes involved in DNA replication (RAD17-RFC and PCNA-CHL12-RFC2-5 complexes) and chromatin modification (E2F-6 complex, STAGA and HMGB1-2 complex; Fig 2A). In addition, we identified the mTORC2 complex and the 20S proteasome. As proteasomal activity is essential for ordered progression through the cell cycle (Koepp *et al*, 1999), and as the proteasome is regulated by mTOR signaling (Zhao *et al*, 2015) and is a potent drug target in human cancers (Goldberg, 2012), we decided to study the proteasome in more detail. The complete 26S proteasome is comprised of the core catalytic 20S proteasome and the regulatory 19S lid. Unexpectedly, all components in the 20S proteasome displayed an increase in $\Delta T_m$ upon palbociclib treatment, whereas the 19S subunits were largely unaffected (Fig 2B).

To understand the physiological impact that palbociclib has on the proteasome, we first measured proteasomal activity in MCF7 cells using a fluorescent peptide substrate Me4BodipyFL-Ahx₃Leu₃VS (Berkers *et al*, 2007). After 1-h treatment, palbociclib activated the degradation of the peptide substrate in a dose-dependent manner (Fig 2C), whereas MG-132, a proteasomal inhibitor, abolished substrate degradation. Another CDK4/6 inhibitor, ribociclib, also increased proteasomal substrate degradation, although to a lesser extent. The proteasomal activation persisted when cells were incubated with palbociclib for 30 h and the level of proteasomal activation with 1 μM palbociclib was comparable to that observed with a

known proteasomal activator, the mTOR inhibitor Torin-1 at 0.1 μM concentration (Fig 2D). Note that 1 μM concentration of palbociclib is lower than the typical range (1.4–4.8 μM) seen in tumor xenografts after treating mice with 10 mg/kg palbociclib (Nguyen *et al*, 2010). Palbociclib also activated the proteasome in T47D and HeLa cells (Fig EV4A and B), indicating that this is not a cell type-specific effect of the drug.

Activity changes observed with peptide substrates do not necessarily reflect changes in proteasomal activity toward protein substrates. We thus examined palbociclib effects on proteasomal activation using the ubiquitin^G76V-GFP biosensor (Dantuma *et al*, 2000). 30-h palbociclib treatment of HeLa cells expressing Ub^G76V-GFP resulted in a major reduction in GFP signal, starting with submicromolar doses, consistent with proteasomal activation (Fig 2E). To verify that palbociclib-induced GFP degradation was specific for protein degradation, and not due to changes in protein synthesis, we treated Ub^G76V-GFP expressing cells simultaneously with palbociclib and protein synthesis inhibitor cycloheximide (CHX) or proteasome inhibitor MG-132. Palbociclib increased protein degradation even in the presence of CHX, but not in the presence of MG-132 (Fig 2F). Increased protein degradation in the presence of CHX was also seen with ribociclib, a structurally similar CDK4/6 inhibitor (Fig 2F).

We also examined whether these results could be explained by increased autophagy, as palbociclib was recently shown to activate autophagy in specific cancer cell lines (Vijayaraghavan *et al*, 2017). However, Western blot analysis of SQSTM1/p62 and LC3A/B protein levels cells did not reveal any significant increase in autophagy in MCF7 to be caused by palbociclib (Fig EV4C and D). Finally, we induced protein aggregation in cells using the proteasome inhibitor MG-132 and observed that palbociclib can enhance the clearance of aggregated proteins in a proteasome-dependent manner when MG-132 is washed out (Fig EV4E).

## Palbociclib induces polyubiquitin chain degradation without activating the whole ubiquitin–proteasome pathway

To validate that palbociclib affects proteasomal degradation globally, we examined the total levels of ubiquitylated cellular proteins by Western blot. This revealed that 1 μM palbociclib treatment reduced the overall level of ubiquitin-conjugated proteins in MCF7 cells (Fig 2G and H) without changing the 20S proteasome levels (Fig 2G). To confirm that the observed reduction in ubiquitin conjugates is related to polyubiquitin chains that are targeted to proteasomal degradation (Kulathu & Komander, 2012), we treated MCF7 cells with 1 μM palbociclib for 10 h in the presence or absence of the proteasome inhibitor bortezomib. We then enriched total polyubiquitin and analyzed the linkages by targeted parallel reaction monitoring (PRM) mass spectrometry (Heap *et al*, 2017). This analysis confirmed that amounts of K48 chains, as well as the K6, K29, and K63 chains, were significantly reduced upon palbociclib treatment, and this reduction was dependent on proteasomal activity (Fig 3). Note that the BRCA1/BARD1 ubiquitin E3 ligase (Kulathu & Komander, 2012) generates K6 chains, and thus, palbociclib-dependent reduction in K6 chains could be relevant for BRCA1 mutant breast cancers.

If palbociclib activates the whole ubiquitin–proteasome pathway, palbociclib treatment should increase the levels of ubiquitin chains when ubiquitin breakdown is blocked by proteasome inhibition.

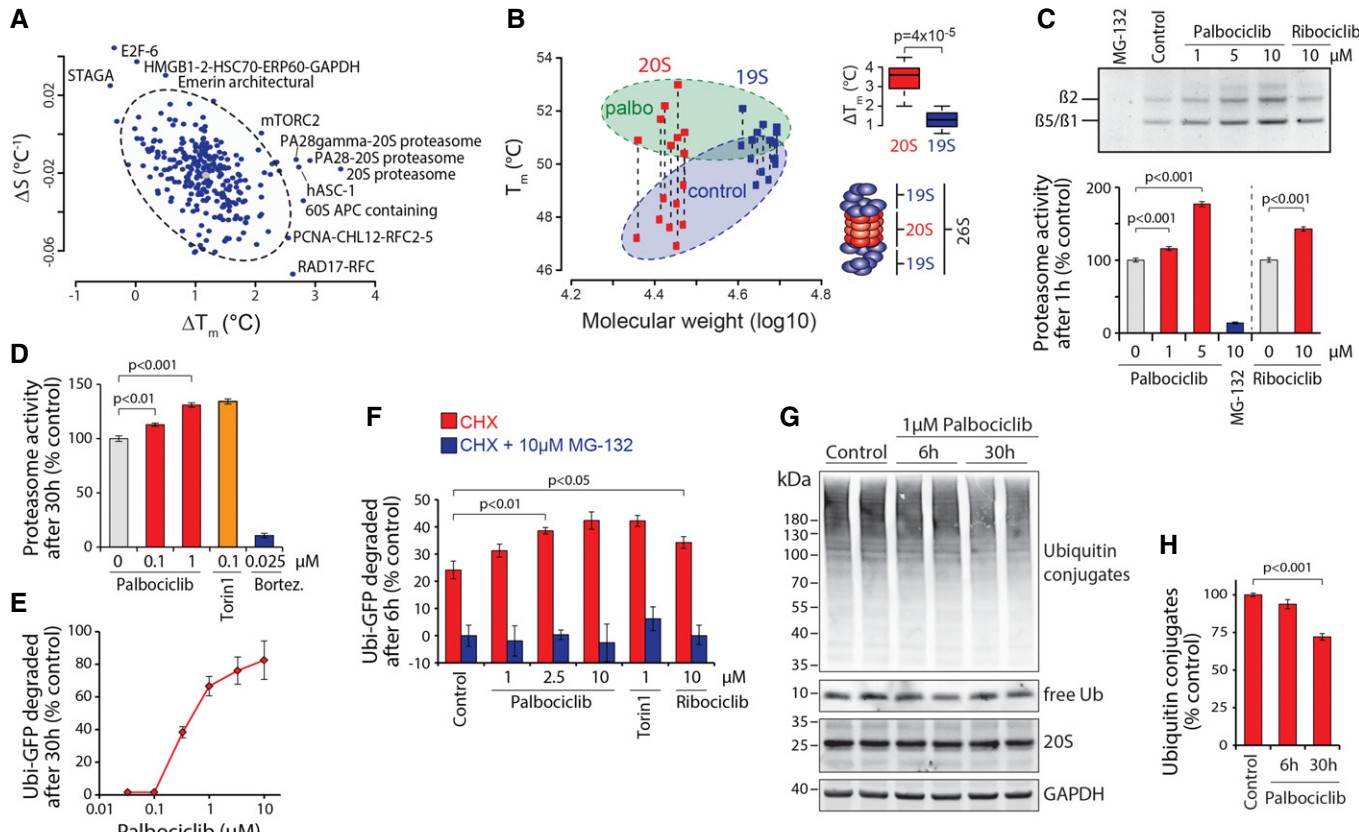

**Figure 2. Palbociclib activates proteasomal protein degradation.**

A   Plot of MCF7 cell protein complexes displaying palbociclib-induced thermal shifts and changes in the slope of the melting curve ($\Delta S$). Circled light gray area represents 95% confidence interval for the protein complexes and the outliers, potential targets of palbociclib are indicated.

B   Melting temperatures of individual components of the 20S (red squares) and 19S (blue squares) proteasome in control (circled blue area) and palbociclib (circled green area)-treated MCF7 cells as a function their molecular weight. Dashed lines connect the corresponding proteasome components in control and palbociclib samples. Insert on the top right displays the $\Delta T_m$ difference between 20S and 19S proteasome subunits. Insert on the bottom right is a schematic representation of the 26S proteasome. The line that divides the box into two parts represents the median. The end of the box shows the upper and lower quartiles. The extreme lines show the highest and lowest value.

C   Proteasome activity in MCF7 cells as measured using Me4BodipyFL-Ahx$_3$Leu$_3$VS probe. Cells were treated with the indicated compounds for 1 h followed by 1 h with the probe. Fluorescence was quantified using flow cytometry ($n = 4$). The top panel displays the fluorescence signal from the probe covalently bound to the proteasome subunits separated on a SDS–PAGE gel.

D   MCF7 cell proteasome activity after 30-h treatment with indicated chemicals as measured by Me4BodipyFL-Ahx$_3$Leu$_3$VS probe ($n = 3–4$).

E   Dose dependency of Ubi$^{G76V}$-GFP degradation in HeLa cells after 30-h palbociclib treatment. GFP levels were quantified using flow cytometry ($n = 3$).

F   Specificity of palbociclib-induced Ub$^{G76V}$-GFP degradation was assessed by treating HeLa cells with the indicated chemicals together with cycloheximide (CHX) for 6 h ($n = 3–4$). The proteasome specificity of GFP degradation was further validated using treatment with 10 μM MG-132.

G   Western blot analysis of ubiquitin-conjugated proteins and 20S proteasomal levels in MCF7 cells.

H   Quantification of ubiquitin-conjugated protein levels from Western blots in (G) ($n = 4$).

Data information: In panels (C, D, E, F, and H), data are presented as means ± SD; each *n* represents an individual biological replicate. *P*-values were determined by two-tailed Student's *t*-test for panel (B) and by ANOVA and two-tailed Student's *t*-test with Holm–Sidak *post hoc* test for panels (C, D, F and H); ns depicts not significant ($P > 0.05$).

However, this was not the case, as none of the ubiquitin chain types displayed increased levels when cells were co-treated with palbociclib and bortezomib (Fig 3). When the experiments were done with the proteasome inhibitor MG-132, similar conclusions were reached (Fig EV5). However, it is worth noticing that bortezomib and MG-132 did not induce fully identical ubiquitin levels, most likely due to the lower specificity of MG-132. Altogether, these data show that palbociclib does not cause increases in protein ubiquitylation, and thus, the palbociclib-induced protein degradation is due to a specific activation of the proteasome. As the rate of degradation for most

proteins is regulated by ubiquitylation, not proteasome activity, the palbociclib-induced proteasomal activation may only affect a specific subset of proteins, such as those with K48/K63 branched linkages (Ohtake *et al*, 2018).

## Palbociclib activates the proteasome through an indirect, ECM29-mediated mechanism

We next asked whether the proteasomal activation is a result of direct interaction between palbociclib and the proteasome or

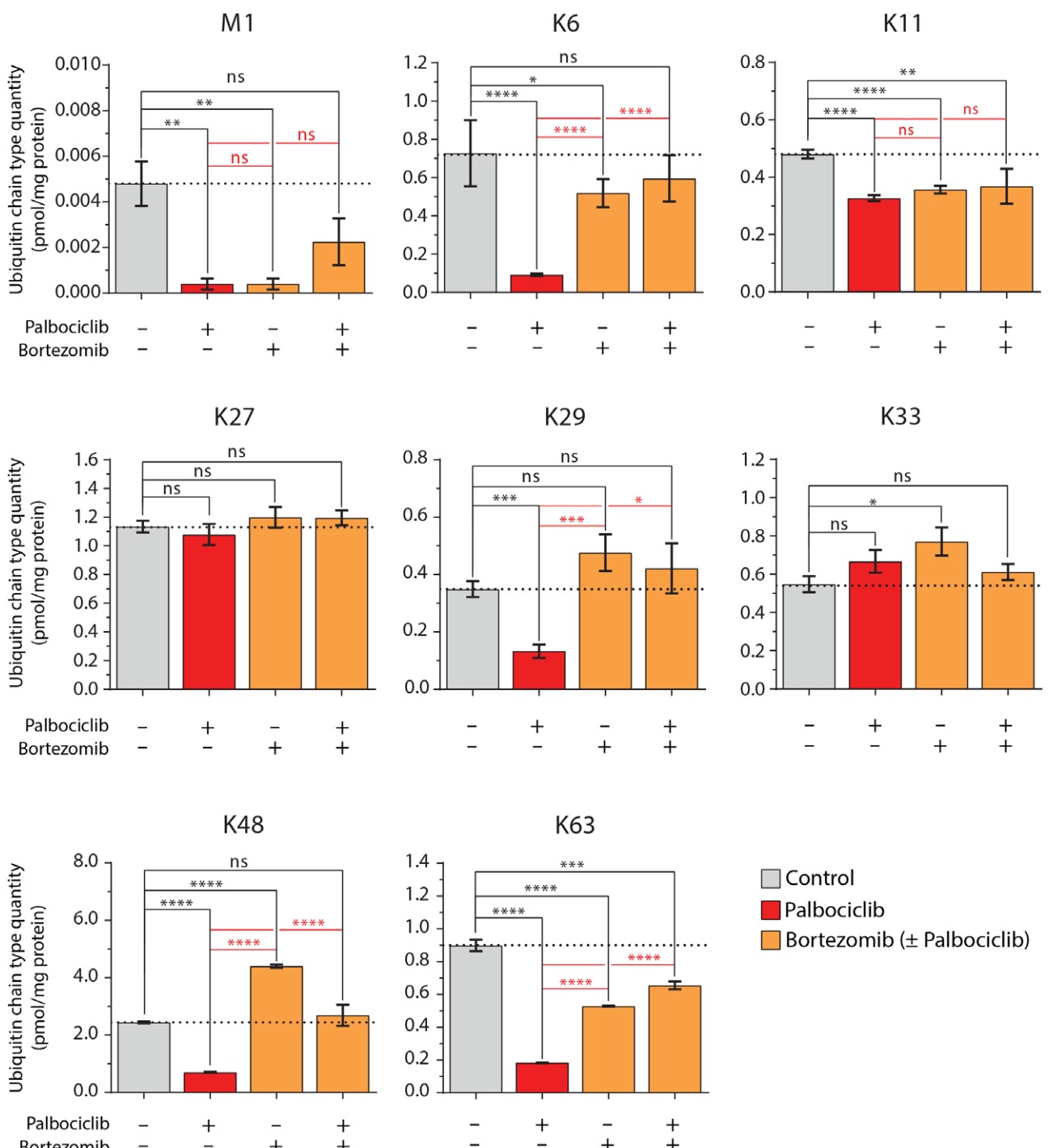

**Figure 3. Palbociclib activates the proteasome largely independently of the ubiquitin system.**

MCF7 cells were treated with 1 μM palbociclib and 100 nM bortezomib for 10 h, and the ubiquitin linkages were absolutely quantified by targeted PRM mass spectrometry. Data show the absolute levels of methionine-1 (M1) and all lysine (K6, K11, K27, K29, K48, and K63) linkages ($n = 3$). Data are presented as means ± SEM; each $n$ represents an individual biological replicate; $P$-values were determined by two-tailed Student's $t$-test; ns depicts not significant ($P > 0.05$); *: $P < 0.05$; **: $P < 0.01$; ***: $P < 0.001$; ****: $P < 0.0001$.

whether it is mediated by other cellular proteins. *In vitro* proteasome activity assays indicated that the increased proteasome activity was not due to a direct palbociclib effect on 20S proteasome (Fig 4A). Next, we examined the primary targets of palbociclib and ribociclib, CDK4 and CDK6. Knockdown of CDK4 induced a negligible increase in proteasome activity in HeLa and MCF7 cells (Fig 4B).

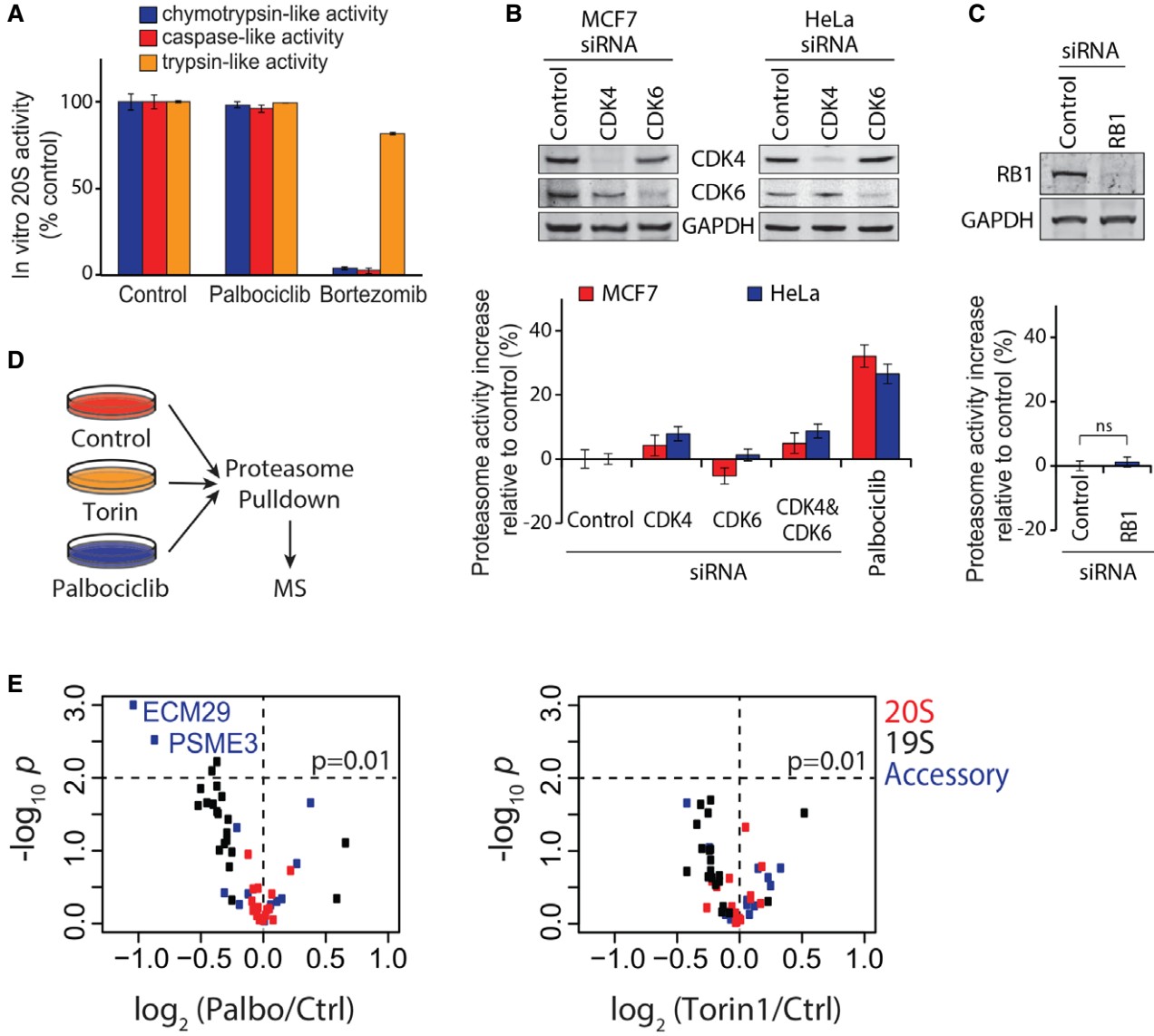

**Figure 4.  Palbociclib activates the proteasome indirectly and reduces the association of ECM29 with the proteasome.**

A   *In vitro* 20S proteasome activity assay with peptide substrates in the presence of 5 μM palbociclib or 10 nM bortezomib shows that palbociclib does not inhibit the proteasome directly (*n* = 3).

B   Proteasome activity levels, as measured by Me4BodipyFL-Ahx₃Leu₃VS probe, after siRNA-mediated CDK4/6 knockdown and palbociclib in HeLa and MCF7 cells (*n* = 3–4). Knockdown efficiency was analyzed by Western blotting. These data indicate that proteasome activation is likely independent of CDK4/6 inhibition through palbociclib.

C   Same as (B), but siRNAs targeted RB1 in HeLa cells (*n* = 3).

D   Workflow schematic of the mass spectrometry-based analysis of proteasome activation mechanism. Palbociclib (10 μM) and Torin-1 (1 μM) treatments of MCF7 cells lasted 4 h (*n* = 3).

E   Volcano plots showing abundance changes and statistical significances for proteasome subunit levels in palbociclib- (left) and Torin-1 (right)-treated MCF7 cells. Proteasomes show significantly reduced levels of ECM29 upon palbociclib treatment.

Data information: In panels (A–C), data are presented as means ± SD; each *n* represents an individual biological replicate. *P*-value for panel (C) was determined by two-tailed Student's *t*-test; ns depicts not significant (*P* > 0.05).
Source data are available online for this figure.

Knockdown of CDK6 had no effect, and combined CDK4 and CDK6 knockdown was similar to CDK4 alone (Fig 4B). Proteasome activation was also independent of RB1, the best-established phosphorylation target of CDK4 and CDK6 (Fig 4C). We also examined whether proteasomal activation by palbociclib occurs at a certain phase of the cell cycle by co-staining MCF7 cells with the fluorescent peptide substrate and a DNA binding dye. Palbociclib increased proteasome activity similarly in G1, S and G2/M phases, indicating cell

cycle-independent activation (Fig EV4F). Altogether, these data suggest that palbociclib affects proteasome activity in a CDK4, CDK6, and cell cycle-independent manner.

Palbociclib and ribociclib have overlapping off-target profiles (Sumi *et al*, 2015), making it possible that previously unappreciated targets mediate the proteasomal activation. Furthermore, as mTOR is a known regulator of proteasome (Zhao *et al*, 2015), the weak inhibition of PI3K/AKT/mTOR signaling could be involved in the proteasome activation. To gain insight into the mechanism of how palbociclib activates the proteasome, we immunoprecipitated proteasomes with anti-α4 20S subunit antibody after 4-h treatments with palbociclib or the mTOR inhibitor Torin-1 and used quantitative mass spectrometry to compare changes in proteasomal protein levels and in phosphorylation sites in proteasomal proteins to untreated controls (Figs 4D and EV6A and Table EV4). The identified phosphosites were largely non-overlapping between palbociclib and Torin-1, suggesting independent mechanisms for proteasome activation (Fig EV6B and Table EV5). Approximately half of the phosphosites in the pulldown from palbociclib-treated cells were proline-directed and thus potential substrates for MAPK and CDKs (Wang *et al*, 2007). The remaining sites were highly enriched for acidic residues after the phosphorylated amino acid residue (Fig EV6C), suggestive of phosphorylation by casein kinase 2, which was identified as a potential palbociclib target (Fig 1E). However, inhibition of casein kinase 2 by quinalizarin (CK2i; 5 μM for 2 h) did not activate the proteasome, but slightly reduced it (Fig EV6C, right). Thus, although palbociclib may affect both casein kinase 2 and PI3K/AKT/mTOR signaling (Figs 1E and EV3), these mechanisms are unlikely to explain the proteasomal activation by palbociclib.

The lack of strong candidate phosphorylation sites in palbociclib-treated cells prompted us to analyze whether palbociclib could mediate its effects on proteasome through protein levels. Palbociclib-induced changes on the protein levels were much larger than those induced by Torin-1 (Fig 4E), suggesting that palbociclib, unlike Torin-1, may affect proteasome structure and/or assembly. While many 19S subunits displayed reduced levels in the presence of palbociclib, the proteasome activator complex subunit 3

(PSME3), and the proteasomal scaffold protein ECM29 had the largest palbociclib-induced decline in abundance (Fig 4E). ECM29, a 204-kDa protein, has been reported to inhibit proteasomal activity in yeast (Finley *et al*, 2016), possibly by disassembling the proteasome (Wang *et al*, 2017). ECM29 binding to the 20S proteasome also changes the proteasome conformation (De La Mota-Peynado *et al*, 2013), which likely explains the increased 20S thermal stability upon ECM29 dissociation from the proteasome.

We proceeded to examine whether palbociclib-induced proteasomal activation could be mediated by ECM29. Consistent with the role of ECM29 as a proteasomal inhibitor (Finley *et al*, 2016), siRNA-mediated depletion of ECM29 increased proteasomal activity in both MCF7 and HeLa cells (Figs 5A and EV7A). Combining ECM29 knockdown with palbociclib treatment did not result in any further increase in proteasomal activity as measured by both the proteasomal activity probe Me4BodipyFL-Ahx$_3$Leu$_3$VS and Ub$^{G76V}$-GFP degradation (Figs 5A and B, and EV7A). Thus, ECM29 is required for the palbociclib-induced proteasomal activity, suggesting that ECM29 mediates the palbociclib-induced proteasome effects.

### ECM29 is critical for normal cell proliferation

Next, we examined the role of ECM29 in the growth and proliferation of breast cancer cells. We induced loss of ECM29 using CRISPR/Cas9 gene editing in MCF7 cells and established a cell line which appeared to be heterozygous knockout of ECM29, as validated by examining protein levels. This cell line displayed extremely slow proliferation (Fig 5C) compared to the parental unaltered cell line. We were not able to create homozygous knockout of ECM29, most likely due to homozygous knockout being lethal for MCF7 cells. Knockdown of ECM29 using siRNA similarly inhibited MCF7 and T47D cell proliferation (Fig EV7B), confirming that ECM29 is involved in cell proliferation. The ECM29 knockout cells were also much larger and many of the cells stained positively for acidic β-galactosidase activity, an established marker for cellular senescence (Fig 5D). We further studied the role of ECM29 in establishing a senescence-like state by examining two other senescence-associated

---

**Figure 5.** **ECM29 mediates palbociclib-induced proteasomal activation and may function as a putative biomarker for palbociclib treatment efficacy in breast cancer.** ▶

A  Proteasome activity, as measured by Me4BodipyFL-Ahx$_3$Leu$_3$VS probe, after siRNA-mediated knockdown of ECM29 and subsequent 6-h treatment with palbociclib (1 μM) or Torin-1 (0.1 μM) in MCF7 cells (*n* = 4). Data show that palbociclib-induced proteasome activation is dependent on ECM29.

B  Proteasomal activity as measured using Ubi$^{G76V}$-GFP degradation. Palbociclib concentration was 1 μM, bortezomib 50 nM (*n* = 3–4).

C  Proliferation of WT and ECM29$^{+/-}$ MCF7 cells (*n* = 3) shows that partial loss of ECM29 reduces proliferation. Western blot levels of ECM29 and GAPDH are shown as insert.

D  β-galactosidase activity staining of wild-type (WT) MCF7 cells and ECM29 heterozygous CRISPR mutant MCF7 cells (ECM29$^{+/-}$) shows increased senescence marker levels in ECM29$^{+/-}$ cells (*n* = 3–4). Note, that much like palbociclib treatment, the partial loss of ECM29 also increases cell size significantly. Scale bar is 100 μm.

E  Western blots of senescence markers p21 and S139 phosphorylated H2AX in MCF7 cells after 48 h of siRNA-mediated knockdown. β-Tubulin was used as a loading control.

F  Quantification of S139 phosphorylated H2AX levels shown in panel (E) (*n* = 3 independent samples from a single experiment).

G  Quantification of p21 protein levels shown in panel (E) (*n* = 3 independent samples from a single experiment).

H  Kaplan–Meier relapse-free survival for all analyzed breast cancer patients with above (high ECM29) or below (low ECM29) median levels of ECM29 mRNA (*n* = 3,554). Hazard ratio and log-rank *P* value are shown at bottom left.

I  Same as (E) but only HER2-/neu-positive breast cancer (*n* = 84 in both groups).

J  Same as (E) but only those with receiving endocrine therapy (*n* = 999).

Data information: In panels (A–D, F, G), data are presented as means ± SD; each *n* represents an individual biological replicate. *P*-values for panels (A and B) were determined by ANOVA and two-tailed Student's *t*-test with Holm–Sidak *post hoc* test; ns depicts not significant (*P* > 0.05); *P*-values for panels (D, F, and G) were determined by two-tailed Student's *t*-test. For panels (H–J), the log-rank *P*-values were provided by the KMplot online service (Gyorffy *et al*, 2010). Source data are available online for this figure.

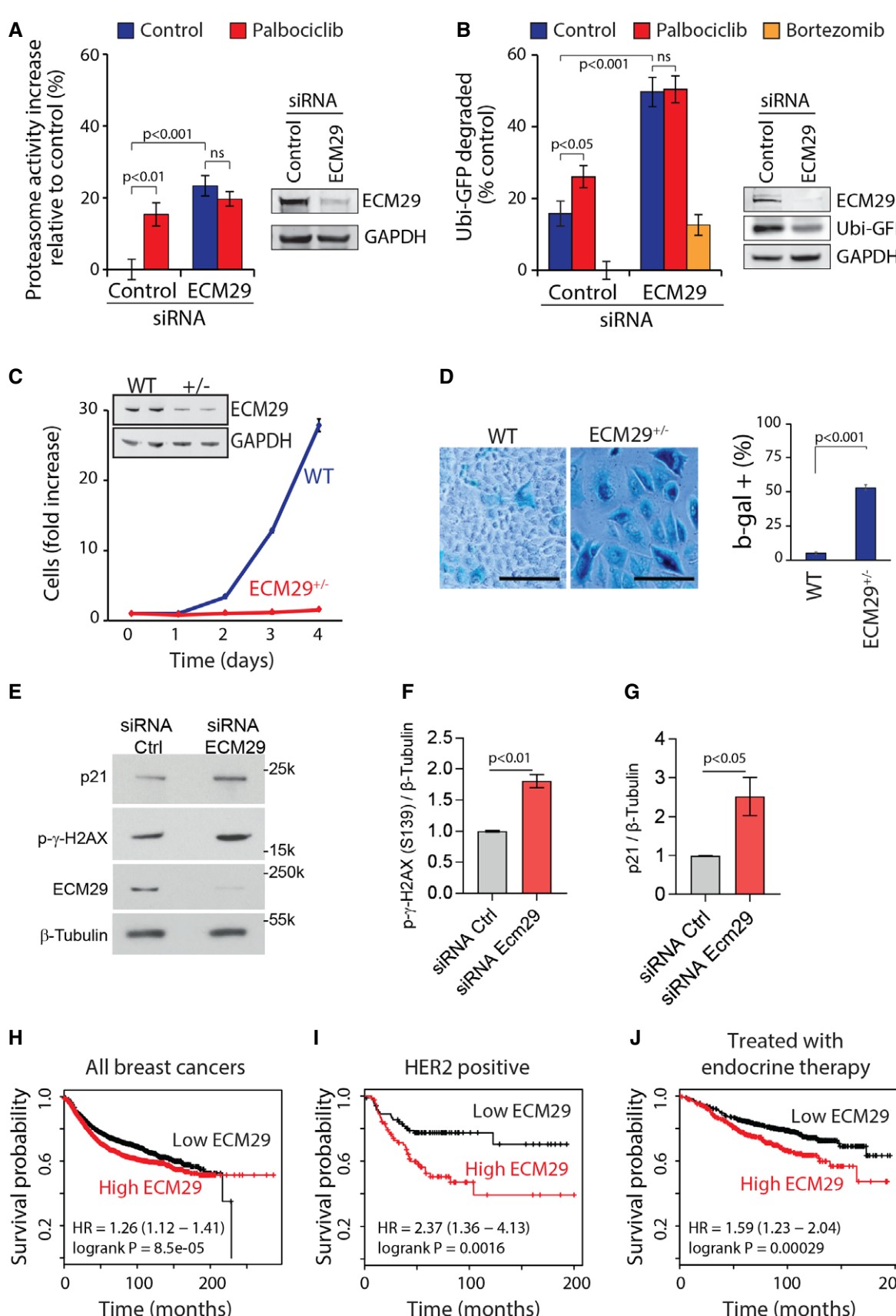

Figure 5.

markers, S139 phosphorylation of Histone H2A.X (γH2AX) and p21 protein levels (Lawless *et al*, 2010), in MCF7 cells where ECM29 was silenced using siRNAs (Fig 5E). Both of these markers displayed marked increase after silencing of EMC29 (Fig 5F and G). This suggests that ECM29, or the associated proteasome activity, may be involved in regulation of cell senescence, a feature previously associated with palbociclib treatment (Anders *et al*, 2011; Vijayaraghavan *et al*, 2017). However, it should be highlighted that we only examined short-term consequences of ECM29 knockdown and the senescence-like phenotype may differ from physiological senescence. Furthermore, our observation that palbociclib increases proteasome activity, but not ubiquitination levels, is consistent with the previous work, suggesting that the presence of ECM29 on the proteasome induces a closed conformation of the substrate entry channel of the core particle (De La Mota-Peynado *et al*, 2013). Altogether, this makes EMC29 a likely mediator of the palbociclib-induced proteasome activation.

### ECM29 expression may predict relapse-free survival in breast cancer

To characterize the potential role of ECM29 as a mediator of palbociclib effects in breast cancer, we examined the potential breast cancer association of ECM29 through the publicly available gene expression data on relapse-free breast cancer patient survival (Gyorffy *et al*, 2010). Lower expression of ECM29 displayed a marginally longer relapse-free survival time when all breast cancer cases were analyzed [hazard ratio (HR) = 1.26, log-rank $P = 8.5 \times 10^{-5}$, $n = 3,554$; Fig 5H]. However, lower expression of ECM29 associated with substantially longer survival times in the patient population with HER2$^+$ cancers (HR = 2.37, $P = 0.016$, $n = 168$; Fig 5I). Consistently, CDK4/6 inhibitors have been shown to be effective against HER2$^+$ cancers in animal models (Goel *et al*, 2016). Furthermore, ECM29 expression levels could predict survival benefit in patients receiving endocrine therapy (HR = 1.59, $P = 0.00029$, $n = 999$; Fig 5J). Additional gene expression datasets provided further support for the correlation between low expression levels of ECM29 and longer breast cancer patient survival (Fig EV7C). Altogether, our data suggest that the expression of ECM29 should be investigated as a potential biomarker to better identify those individuals benefiting from palbociclib alone or in combination with endocrine therapy.

### Proteasomal activity is required for palbociclib-induced cellular phenotypes

To gain insight into how increased proteasomal activity could limit breast cancer growth, we examined palbociclib-induced degradation of endogenous proteins in MCF7 cells using Western blotting. A 6-h treatment with palbociclib-induced proteasome-dependent degradation of 3-hydroxy-3-methylglutaryl-CoA reductase (HMGCR; Fig 6A), the rate-limiting enzyme of the mevalonate pathway, which is critical for normal cell growth and proliferation (Miettinen & Bjorklund, 2015). Levels of housekeeping proteins GAPDH and SOD1 were not affected by palbociclib. At a later timepoint (30 h), the levels of CDK4 were also reduced by palbociclib in a proteasome-dependent manner (Fig 6A and B), suggesting that in addition to the direct inhibition of CDK4/6 kinase activity, long-term

treatment with palbociclib may also inhibit cell proliferation by a positive feedback loop resulting from proteasomal CDK4 degradation.

To understand the role of increased proteasome activity in palbociclib-induced cell cycle arrest, we isolated MCF7 and T47D cells in G1 phase using centrifugal elutriation and treated these cells with palbociclib, bortezomib, or both (Fig 6C). Note that we used bortezomib concentrations that do not completely block all proteasomal activity so that cell viability would be maintained (data not shown). Cell cycle analysis showed that palbociclib prevented progression beyond the G1 phase and bortezomib alone induced cell accumulation at G2/M. Unexpectedly, bortezomib, which counteracts the proteasomal activation by palbociclib, could override the palbociclib-induced G1 arrest in both cell lines (Fig 6D and E). Thus, these data suggest that proteasomal activity is required for palbociclib-induced cell cycle arrest. Increased proteasomal activation by palbociclib may be an additional mechanism to ensure the completeness of the G1 arrest.

If the proteasomal activation by palbociclib is involved in some of the cellular phenotypes caused by palbociclib treatment, then resistance to palbociclib should also be seen in the form of lower proteasomal activity. Recently, MCF7 and TD47 cell lines resistant to CDK4 and CDK6 inhibitors have been created (Jansen *et al*, 2017), and these cells display normal cell cycle profile and proliferation rate in the presence of palbociclib and ribociclib. We examined the proteasomal activity in these cells and observed that the CDK4/6 inhibitor-resistant cells displayed much lower baseline proteasome activity than the wild-type cells (Fig 6F). Furthermore, when treated with 1 μM palbociclib for 6 h, the resistant cells displayed much smaller increase in proteasome activity, even after normalization for the baseline difference (Fig 6G). As the resistant cell lines were obtained by selecting for cells capable of proliferation in the presence of ribociclib (Jansen *et al*, 2017), these results link the cell cycle responses of palbociclib/ribociclib to the proteasomal activity of the cell.

Palbociclib has been reported to induce a senescence-like state in multiple cell types (Anders *et al*, 2011), and at least in MCF7 cells, this senescence-like state is dependent on seemingly unrelated cellular components, like integrin β3 and TGF beta pathway (Rapisarda *et al*, 2017). We examined the effects of modest proteasome inhibition by bortezomib on the palbociclib-induced senescence-like state in MCF7 cells. Similarly to the partial loss of ECM29 (Fig 5D), palbociclib (1 μM) treatment significantly increased MCF7 cell size and the activity of senescence-associated β-galactosidase, while decreasing the levels of Ki67 (Fig 7A–C, E and F), all common features of senescence. However, all of these effects were largely or completely suppressed by simultaneous treatment with bortezomib (7.5 nM). We also examined the levels of Ser139 phosphorylated Histone H2A.X (γH2AX), which is a marker of cellular senescence and DNA damage (Lawless *et al*, 2010). γH2AX was upregulated by palbociclib treatment, and this effect was dependent on proteasomal activity (Fig 7A and D). While this is consistent with the other senescence markers, it also suggests that palbociclib may have a role in DNA damage accumulation and/or repair, especially as our MS-CeTSA results identified RAD17-RFC and PCNA-CHL12-RFC2-5, which are protein complexes responding to DNA damage as a potential palbociclib targets. Altogether, these results indicate that the proteasomal

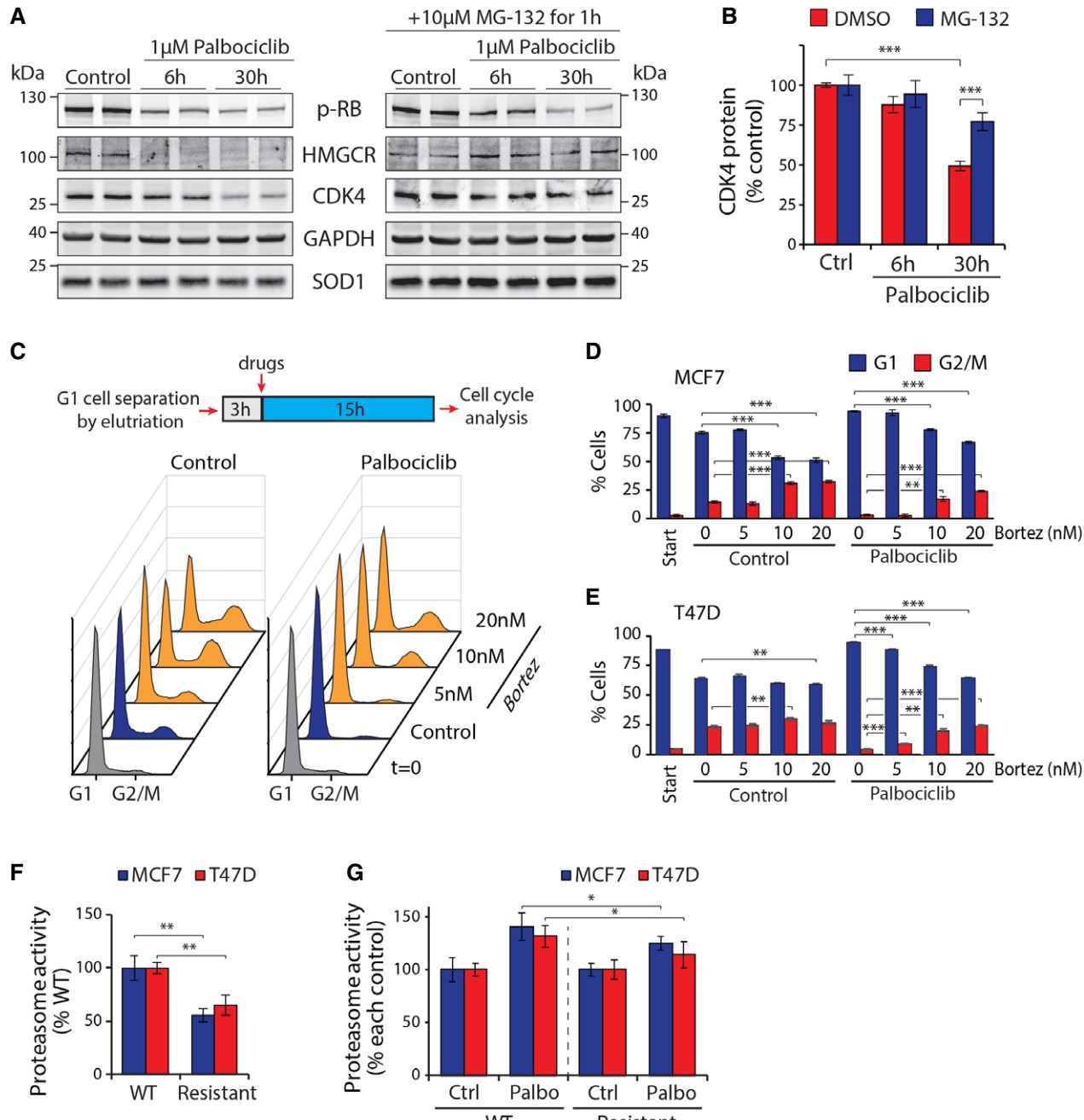

**Figure 6. High proteasomal activity is involved in the palbociclib-induced G1 arrest.**

A MCF7 cells were treated with 1 μM palbociclib for 0, 6, or 30 h to examine levels of endogenous proteins in response to palbociclib-induced protein degradation. Additionally, a second set of cells were additionally treated with 10 μM MG-132 for the last 1 h before sample collection to inhibit the proteasome. HMGCR and CDK4 protein levels are reduced by palbociclib treatment.

B Protein levels of CDK4 were quantified from the Western blot samples ($n = 4$).

C Top, Workflow schematic of cell cycle synchronization and analysis of cell cycle progression. Bottom, Representative DNA histograms of each sample at the time of analysis from MCF7 cells. Palbociclib prevents progression beyond the G1 phase, which can be overridden by addition of bortezomib.

D, E Quantification of cell cycle status after G1 cell cycle synchronization and 15 h with 1 μM palbociclib with or without bortezomib treatment in MCF7 cells (D) ($n = 4$) and T47D cells (E) ($n = 3$).

F Proteasome activity levels, as measured by Me4BodipyFL-Ahx₃Leu₃VS probe, in wild-type (WT) and palbociclib-resistant (Resistant) MCF7 and T47D cells ($n = 4$).

G Same as (F), but data are from MCF7 and T47D cells after 6-h treatment with 1 μM palbociclib. The data have been normalized to each control to enable comparison of the relative proteasome activation caused by palbociclib ($n = 4$).

Data information: In panels (B, D–G), data are presented as means ± SD; each *n* represents an individual biological replicate. *P*-values were determined by ANOVA and two-tailed Student's *t*-test with Holm–Sidak *post hoc* test; ns depicts not significant ($P > 0.05$); *: $P < 0.05$; **: $P < 0.01$; ***: $P < 0.001$.
Source data are available online for this figure.

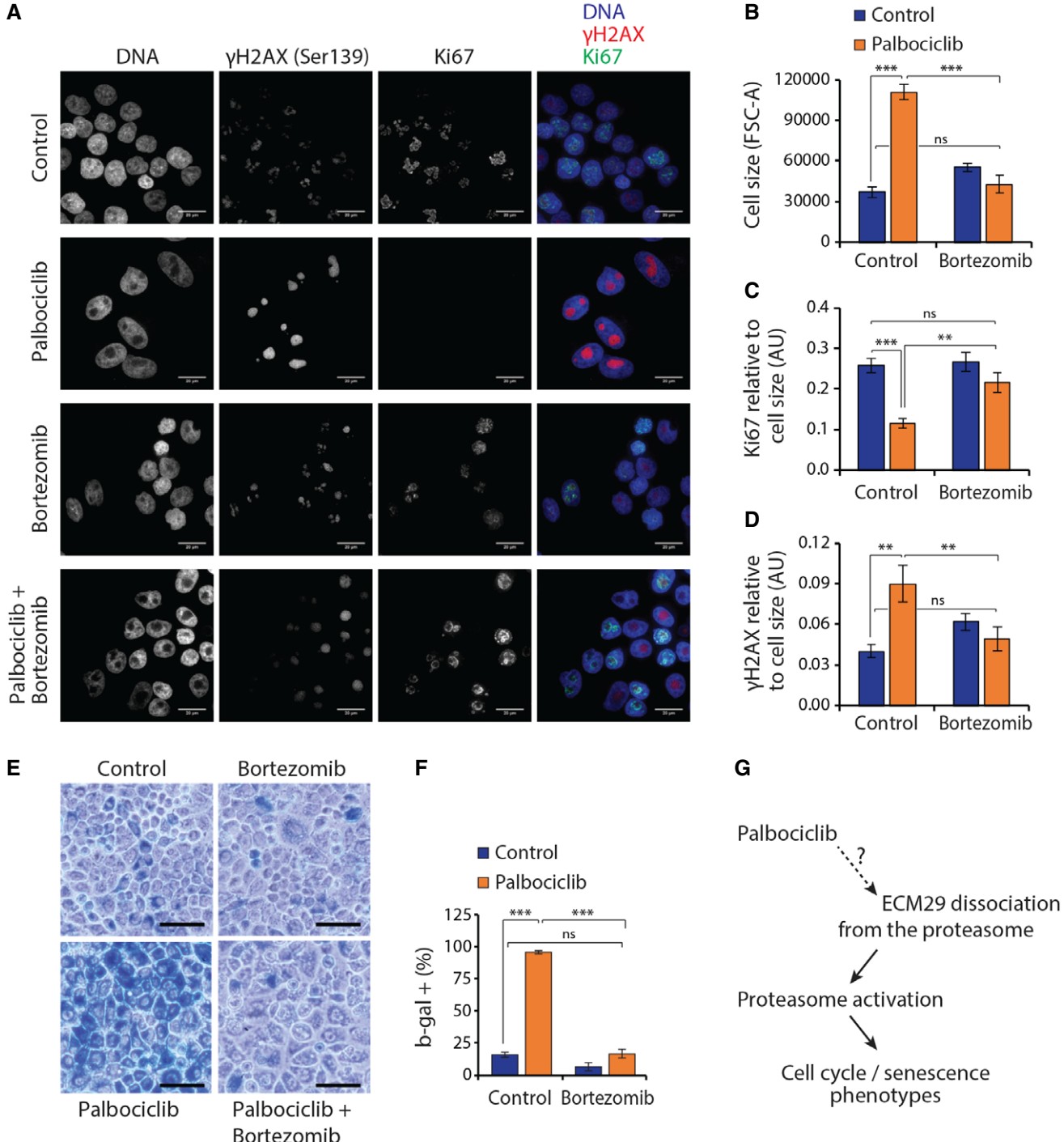

**Figure 7. Proteasomal inhibition suppresses palbociclib-induced senescence phenotype.**

A    Representative maximum-intensity projections of MCF7 cells treated with the 1 µM palbociclib and/or 7.5 nM bortezomib. Note that 7.5 nM bortezomib inhibits proteasome only partially. Fixed and permeabilized cells were stained with Alexa Fluor 488-conjugated Ki67 antibody, Alexa Fluor 647-conjugated phospho-Histone H2A.X (γH2AX) (pSer139) antibody, and DAPI. All images were acquired with the same magnification, and scale bar is 20 µm.

B–D    Flow cytometry-based quantifications of cell size (B), Ki67 levels (C), and pSer139 γH2AX levels (D) from the samples presented in panel (A) ($n = 4$).

E    Senescence-associated beta-galactosidase activity (blue staining) in MCF7 cells untreated (control) or treated with 7.5 nM bortezomib, 1 µM palbociclib, or a combination of both. Scale bar is 50 µm.

F    Quantification of the β-galactosidase activity levels is shown in panel (E) ($n = 4$).

G    Schematic model of palbociclib action on cell cycle and cell senescence.

Data information: In panels (B–D and F), data are presented as means ± SD; each $n$ represents an individual biological replicate. $P$-values were determined by ANOVA and two-tailed Student's $t$-test with Holm–Sidak post hoc test; ns depicts not significant ($P > 0.05$); **: $P < 0.01$; ***: $P < 0.001$.

activation by palbociclib is critical for the induction of a senescence-like state, suggesting that the proteasome activation by palbociclib is part of the drug's mode of action.

## Discussion

Proteasomal activity must be tightly regulated to maintain cellular homeostasis. Proteasomal degradation of cyclins and many other cell cycle regulatory proteins is carefully controlled in order to maintain normal cell cycle progression (Koepp *et al*, 1999). Consequently, both activators of proteasome (rapamycin analogs) and direct proteasomal inhibitors (bortezomib) have been considered as treatment options in breast cancer. Using the thermal proteome profiling approach, we identified the proteasome as a novel downstream target for the CDK4/6 inhibitor palbociclib. Further functional analyses revealed that palbociclib causes proteasomal activation at least partly by reducing the proteasomal association of EMC29, which normally suppresses proteasome activity. Similar, albeit less potent, proteasome activation was also observed with ribociclib, a structurally related CDK4/6 inhibitor. Further studies are required to identify the direct targets of palbociclib that mediate the proteasomal disassociation of EMC29, and our thermal proteome profiling datasets are likely to be useful in this. Curiously, the proteasomal activation by palbociclib was not associated with increased protein ubiquitylation, and the palbociclib treatment led to a decrease in several ubiquitin chain types in cells. Nonetheless, palbociclib treatment increased proteolysis, including degradation of endogenous proliferation-related proteins such as HMGCR and CDK4. Consistently, proteasome activation by palbociclib seems to be involved in the reduction in cancer cell proliferation and, especially, in the induction of a senescence-like state (Fig 7G). Overall, our data, together with a recently published "atlas for drug interactions" (Klaeger *et al*, 2017), suggest that palbociclib has additional targets with functional consequences.

Palbociclib has been a highly effective compound in the treatment of patients with advanced ER$^+$ and HER2$^-$ breast cancer (Finn *et al*, 2015; Cristofanilli *et al*, 2016). Given that breast cancers constitute up to 15% of all cancer cases and more than 70% of breast cancers are ER$^+$ and HER2$^-$, palbociclib potentially targets the largest population of cancer patients. A major need to be addressed is the identification of the most responsive patient subpopulation. Although further studies are required, analysis of existing mRNA expression data related to breast cancer survival suggests that ECM29 could be a useful biomarker for HER$^+$ cancers and those patients receiving endocrine therapy. We observed that palbociclib reduces the proteasomal association of ECM29 which causes a major reduction in cell proliferation. Palbociclib could therefore be more beneficial in the population with high ECM29 levels, which predict poorer relapse-free survival in patients receiving endocrine therapy.

Recently, proteasomal inhibition by DYRK2 kinase knockout was shown to reduce cell cycle progression (Guo *et al*, 2016). However, we find that high proteasomal activity associates with G1 arrest induced by CDK4/6 inhibition, which may lead to senescence. Our finding is consistent with earlier observations in other model systems where inhibition of the ubiquitin–proteasome system may

override CDK4/6-mediated G1 arrest (Zhang *et al*, 2002; The *et al*, 2015). Overall, the current and previous works indicate that the role of the proteasome in cell cycle progression is complex, likely context dependent and, importantly, exploited by cancer cells to promote cell proliferation. Instead of a simplistic view where more proteasomal activity is always better for the cell, it seems that appropriate proteasomal homeostasis is required for the maintenance of cell proliferation. Proteolytic homeostasis that supports proliferation and cancer growth may also be dependent on autophagy, inhibition of which was recently reported to synergize with palbociclib (Vijayaraghavan *et al*, 2017). Preclinical data suggest that palbociclib sensitizes myeloma cells to bortezomib-induced cell death (Menu *et al*, 2008), but our work raises the question of whether proteasome inhibition and palbociclib are a good combination for breast cancer therapy due to their partly antagonist activity. Indeed, proteasomal activators may have therapeutic potential in certain cancers and this is worthy of further preclinical and translational investigation.

## Materials and Methods

### Cell culture, elutriations, and transfections

MCF7 and T47D cells were cultured in RPMI and HeLa-TREx cells were cultured in high-glucose DMEM. Media were supplemented with 10% FBS (Sigma-Aldrich), 1% L-glutamine, and 1% penicillin and streptomycin and for T47D additionally with 10 μg/ml human insulin (Sigma-Aldrich). Cells were checked for mycoplasma infections. All experiments were performed in non-confluent cells. For CRISPR/Cas9 knockout, ECM29 was targeted using a dual cleavage strategy with paired guide sequences CCTGAGACTCGACTTGC TATTCA and GGTTGGAGCGTATAGTACTTTGG cloned into the Cas9 D10A vector pX335 and pBABED puro U6. MCF7 cells were transiently transfected with Cas9 and the targeting vectors using Fugene HD. Twenty-four hours post-transfection, edited cells were enriched by culturing them for 48 h in the presence of 2 μg/ml puromycin. Individual cell clones were recovered by limiting dilution in the absence of puromycin, and ECM29 protein levels were assessed by Western blotting.

For cell counting and fluorescence measurements, Accuri C6 cytometer (Becton-Dickinson) was used so that only cells of viable size were included in the analysis as estimated using FSC-A and SSC-A values. All fluorescence values were normalized to cell size (FSC-A), as prolonged incubations with, for example, palbociclib increased the cell size.

siRNAs were obtained from Integrated DNA Technologies. Two independent siRNAs (20 nM each) were combined to target each gene, and the siRNAs were transfected using Qiagen's HiPerFect transfection reagent. The following siRNAs were used: CDK4 (HSC.RNAI.N000075.12.1 and HSC.RNAI.N000075.12.2), CDK6 (HSC.RNAI.N001145306.12.1 and HSC.RNAI.N001145306.12.2), RB1 (HSC.RNAI.N000321.12.1 and HSC.RNAI.N000321.12.2), and ECM29 (HSC.RNAI.N001080398.12.1 and HSC.RNAI.N001080398.12.2). Ub$^{G76V}$-GFP was transfected using Promega's FuGENE transfection reagent. For proliferation assay after ECM29 RNAi in MCF7 and T47D cell culture, cells were cultured in complete medium and absolute cell count was measured every 24 h for 4 days.

MCF7 and T47D cells were synchronized to early/mid G1 using centrifugal elutriation, which was carried out as before (Miettinen & Bjorklund, 2015). Cells were then plated for 3 h after which indicated chemicals were added for 15 h. Cell cycle distributions were analyzed by propidium iodide staining of cell fixed in 70% EtOH and treated with RNAse I.

Proteins and plasmids generated at the University of Dundee for the present study are available to request on the https://mrcppureagents.dundee.ac.uk/ website.

### Antibodies and Western blotting

For Western blots, cells were directly lysed in Laemmli buffer. Validations of MS-CeTSA were carried out by loading equal volume of each temperature fraction on a SDS–PAGE gel. SOD1 band intensities were used to verify equal sample loading. The following antibodies were purchased from Cell Signaling Technology: AKT (#4691), p-AKT S473 (#9271), CDK4 (#2906), CDK6 (#3136), CDK7 (#2916), GAPDH (#5174), p21 (#2947), p70 S6K, p-p70-S6K T389 (#9234), RB (#9309), p-RB S780 (#8180), S6-RP (#2317), p-RPS6 S235/236 (#4858). The S20 Proteasome antibody was from VIVA Bioscience (VB2452), SOD1 antibody was from Sigma-Aldrich (HPA001401), the Ubiquitin antibody was from eBioscience (eBioP4D1), g-H2AX (#05-636) were from Millipore, and the HMGCR and ECM29 antibodies were from Abcam (ab98018 and ab28666, respectively). For the Pathscan Intracellular Signaling array (Cell Signaling Technology), T47D cells were incubated with 0, 0.1, 1, and 10 μM palbociclib for 1 h. Samples were processed as per manufacturer's instructions for fluorescent detection. In all assays, antibodies were used at the concentrations recommended by the supplier and detected using infrared dye-conjugated secondary antibodies and a LICOR Odyssey detection system. Protein levels were quantified using ImageJ, and both GAPDH and SOD1 levels were used to control equal sample loading in Western blots.

### Thermal profiling sample preparation

Sample preparation was performed similar to the previous protocols (Miettinen & Bjorklund, 2014; Savitski *et al*, 2014) with the following modifications. MCF7 cells were trypsinized, washed with PBS, and suspended in PBS supplemented with a protease inhibitor cocktail (Sigma-Aldrich). Suspended cells ($2 \times 10^4$ cells/μl) were treated with 10 μM palbociclib or $H_2O$ for 1 h at 37°C with gentle mixing. Each sample was then separated into 10 fractions, each with $5 \times 10^6$ cells for thermal profiling. Fractions were heated at the indicated temperatures (37–65°C) for 3 min using an Eppendorf Thermomixer with mixing (500 rpm). Fractions were then incubated for 3 min at room temperature and frozen on dry ice. Samples were lysed with four freeze–thaw cycles using dry ice and 35°C water bath. Cell lysates were centrifuged at $16,000 \times g$ for 15 min at 4°C to separate protein aggregates from soluble proteins. Supernatants were collected and used for mass spectrometry and Western blots.

### Proteasome activity assays

Live cell proteasome activity was assessed by pretreating cells with the indicated chemicals followed by 1-h incubation in the presence of 0.5 μM proteasome activity probe (Me4BodipyFL-Ahx$_3$Leu$_3$VS;

BostonBiochem; Berkers *et al*, 2007). After incubation, cells were washed with PBS and analyzed by flow cytometry. The specificity of the proteasome activity probe was also validated on a SDS–PAGE gel, where an equal amount of each sample was loaded to assess proteasome staining.

GFP degradation was measured by reverse transfecting HeLa-TREx cells with the short-lived Ub$^{G76V}$-GFP (Dantuma *et al*, 2000), 12 h later treating the cells with doxycycline to induce GFP expression and another 12 h later adding the indicated chemicals. GFP levels were measured using flow cytometry, and the protein degradation % was calculated by normalizing the data to control (0% degraded) and negative control without the GFP construct (100% degraded). When examining GFP levels in the absence of protein synthesis, cells were treated with 100 μM cycloheximide at the same time as when chemicals of interest were added.

Proteasome activity toward protein aggregates was also measured using the ProteoStat Aggresome detection reagent (Enzo Life Sciences). MCF7 cells were first treated with control or palbociclib and cultured in the presence of MG-132 for 18 h to accumulate protein aggregates. The cells were then washed twice and treated with palbociclib or control. After 4 h, the protein aggregate levels were measured using flow cytometer according to the supplier's instructions. Data were normalized to that 0% represents the basal level of protein aggregates in a control untreated with MG-132 and that 100% represents the level to which MG-132 increased the protein aggregates.

20S proteasome activity *in vitro* was assayed as previously described (Cui *et al*, 2014). Briefly, MCF7 cells were lysed in assay buffer [25 mM HEPES-KOH pH 7.4, 0.5 mM EDTA, 0.05% (v/v) NP-40, 0.001% (w/v) SDS]. 40 μg clarified lysate was preincubated with the inhibitors for 10 min in assay buffer containing 100 μM ATP before the addition of chymotrypsin-like substrate Suc-LLVY-AMC (Enzo Life Sciences), caspaselike substrate Z-Leu-Leu-Glu-AMC (Enzo Life Sciences), or trypsinlike substrate Z-ARR-AMC (Millipore) to 100 μM final concentration. Final reaction volume was 100 μl.

### Cell senescence assays and proliferative markers detection

MCF7 cells were cultured with either 7.5 nM bortezomib, 1 μM palbociclib, or both for 10 days, with media change every second day. Note that higher concentrations of bortezomib became toxic to the cells in long term (data not shown). For β-galactosidase activity, cells were fixed with 4% paraformaldehyde and stained with 0.1% β-gal, 5 mM potassium ferrocyanide, 5 mM potassium ferricyanide, 150 mM NaCl, and 2 mM $MgCl_2$ in 40 mM citric acid/sodium phosphate solution, pH 6.0 for overnight at 37°C. The cells were washed with PBS and imaged using Nikon Eclipse TS100 microscope. For Ki67 and phospho-Histone H2A.X (Ser139) assays, cells were fixed with 4% paraformaldehyde, permeabilized with 0.5% Triton X-100 for 5 min, washed twice with 5% BSA in PBS, and stained with antibodies o/n in +4°C. Ki67 (D3B5) Alexa Fluor 488 conjugate antibody and phospho-Histone H2A.X (Ser139; 20E3) Alexa Fluor 647 conjugate antibody were obtained from CST (#11880 and #9720, respectively). The following day cells were washed twice with 5% BSA in PBS, and antibody staining levels were analyzed using LSR II flow cytometer from BD Biosciences. The antibody staining was validated using microscopy, for which

samples were stained as before, with the exception that the DNA was stained with DAPI after antibody staining, which was followed by two washes with PBS. Cells were imaged with DeltaVision widefield deconvolution microscope in VectaShield mounting media (VectorLabs) using a 60× objective.

### Immunoprecipitation of proteasomes

MCF7 cells cultured in 10-cm dishes were treated for 4 h with 10 μM palbociclib or 1 μM Torin-1 in triplicates or left untreated. Cells were lysed in 25 mM Hepes, pH 7.4, 10% glycerol, 5 mM MgCl$_2$, 1 mM ATP, 1 mM DTT and phosphatase inhibitor cocktail. Clarified lysates were incubated with agarose-immobilized 20S α4 subunit monoclonal antibody (BML-PW9005-0500, Enzo Life Sciences) for 1 h at +4°C, and the beads washed three times with 500 μl lysis buffer without inhibitors.

### Protein extraction and digestion

Protein concentration of samples was determined (BCA Protein Assay) and used to normalize the volume corresponding to the protein quantity necessary for tandem mass tag labeling (100 μg). Final concentration of 50 mM tris (2-carboxyethyl) phosphine (TCEP), 50 mM triethylammonium bicarbonate pH 8.0 in 10% 2,2,2-trifluoroethanol (TFE) was added to protein lysates and heated at 55°C for 30 min. Cysteines were then alkylated in 10 mM iodoacetamide (Sigma), and excess reagent was quenched with 10 mM dithiothreitol (Sigma) and digested overnight at 37°C by adding porcine trypsin (1:50, w/w) (Pierce). Peptides were desalted via C18 Macro SpinColumns (Harvard Apparatus) and dried under vacuum centrifugation.

### Tandem mass tagging labeling

Isobaric labeling of peptides was performed using the 10-plex tandem mass tag (TMT) reagents (Thermo Scientific). TMT reagents (0.8 mg) were dissolved in 40 μl of acetonitrile, and 20 μl was added to the corresponding fractions, previously dissolved in 100 μl of 50 mM triethylammonium bicarbonate, pH 8.0. The reaction was quenched by addition of 8 μl of 5% hydroxylamine after 1-h incubation at room temperature. According to the different temperature points, labeled peptides were combined, acidified with 200 μl of 0.1% TFA (pH ~2) and concentrated using the C18 SPE on Sep-Pak cartridges (Waters).

### Hydrophilic Strong Anion Exchange (hSAX) chromatography

TMT-labeled peptides were subjected to hydrophilic Strong Anion Exchange (hSAX) fractionation (Ritorto *et al*, 2013). Labeled peptides were solubilized in 20 mM Tris–HCl, pH 10.0 and separated on a Dionex RFIC IonPac AS24 column (IonPac series, 2 × 250 mm, 2,000 Å pore size; Thermo Scientific). Using a DGP-3600BM pump system equipped with a SRD-3600 degasser (Thermo Scientific) a 30 min gradient length from 8 to 80% of 1 M NaCl in 20 mM Tris–HCl, pH 10 (flow rate of 0.25 ml/min), separated the peptide mixtures into a total of 34 fractions. The 34 fractions were merged into 15 samples, acidified with 1% trifluoroacetic acid (TFA; pH ~2), desalted via C18 Macro SpinColumns

(Harvard Apparatus), dried under vacuum centrifugation and resuspended in 2% acetonitrile (ACN)/0.1% TFA for LC-MS/MS analysis.

### Liquid chromatography–mass spectrometry (LC-MS)

Peptide samples were separated on an Ultimate 3000 Rapid Separation LC Systems chromatography (Thermo Scientific) with a C18 PepMap, serving as a trapping column (2 cm × 100 μm ID, PepMap C18, 5 μm particles, 100 Å pore size) followed by a 50-cm EASY-Spray column (50 cm × 75 μm ID, PepMap C18, 2 μm particles, 100 Å pore size) (Thermo Scientific) with a linear gradient consisting of 2.4–28% [ACN, 0.1% formic acid (FA)] over 150 min at 300 nl/min. Mass spectrometric identification and quantification were performed on an Orbitrap Fusion Tribrid mass spectrometer (Thermo Scientific) operated in data-dependent, positive ion mode. FullScan spectra were acquired in a range from 400 to 1,500 m/z, at a resolution of 120,000, with an automated gain control (AGC) of 300,000 ions and a maximum injection time of 50 ms. The 12 most intense precursor ions were isolated with a quadrupole mass filter width of 1.6 m/z, and CID fragmentation was performed in one-step collision energy of 32% and 0.25 activation Q. Detection of MS/MS fragments was acquired in the linear ion trap in a rapid mode with an AGC target of 10,000 ions and a maximum injection time of 40 ms. Quantitative analysis of TMT-tagged peptides was performed using FTMS3 acquisition in the Orbitrap mass analyzer operated at 60,000 resolution, with an AGC target of 100,000 ions and a max injection time of 120 ms. Higher-energy C-trap dissociation (HCD fragmentation) on MS/MS fragments was performed in one-step collision energy of 55% to ensure maximal TMT reporter ion yield and synchronous precursor selection (SPS) was enabled to include 10 MS/MS fragment ions in the FTMS3 scan.

### Data processing and quantitative data analysis

Protein identification and quantification were performed using MaxQuant Version 1.5.1.7 (Cox & Mann, 2008) with the following parameters: stable modification carbamidomethyl (C); variable modifications oxidation (M), acetylation (protein N terminus), deamidation (NQ), hydroxyproline (P), quantitation labels with 10 plex TMT on N-terminal or lysine with a reporter mass tolerance of 0.01 Da and trypsin as enzyme with two missed cleavages. Search was conducted using the Uniprot-Trembl Human database (42,096 entries, downloaded March 17, 2015), including common contaminants. Mass accuracy was set to 4.5 ppm for precursor ions and 0.5 Da for ion trap MS/MS data. Identifications were filtered at a 1% false-discovery rate (FDR) at the protein level, accepting a minimum peptide length of 5 amino acids. Quantification of identified proteins referred to razor and unique peptides, and required a minimum ratio count of 2. Normalized ratios were extracted for each protein/conditions and were used for downstream analyses.

 Protein complexes were identified based on annotations in the Corum database (http://mips.helmholtz-muenchen.de/genre/proj/corum/) using the Core Complexes dataset. Only complexes with four or more proteins were included in the analysis. Mean $\Delta T_m$ and $\Delta S$ for each complex was used for plotting the average change

of the complex. Outlier complexes were identified by plotting a 95% confidence interval based on normal-probability contour over the two-dimensional scatter plot data using the *dataEllipse* function in R.

### Data normalization

Stable proteins were classified as those with increasing relative abundance at higher temperatures in control samples. Preliminary analyses showed that those proteins with an average of at least five times higher abundance at 59°C compared to 37°C and which were identified by at least five unique peptides minimized the standard error in denaturation curve fitting. Keratins were omitted from the list of stable proteins as they could be a contamination from sample handling. The 59°C data point rather than any higher temperature point was used as the identification of stable proteins as it had the minimum standard error of the curve fitting. Mean abundance for the identified 32 stable proteins were calculated for each temperature, and the relative abundancies of the proteins at each temperature were divided with the mean abundance for the 32 proteins at the same temperature. For immunoprecipitated proteasomes, protein abundance data were normalized to the total abundance of 20S subunits, excluding the less abundant immunoproteasome subunits.

### Denaturation curve fitting

For the 5,515 proteins quantified by mass spectrometry, only those identified with two to three replicates in both control and palbociclib-treated cells were included for the final analysis. The sigmoid curve from Savitski *et al* (2014),

$$f(T) = \frac{1 - f_0}{1 + e^{-(a/T - b)}} + f_0,$$

where $f_0$ is the plateau at temperature $T \to \infty$, while $a$ and $b$ control the shape of the function. The melting temperature is defined as $f(T_m) = 1/2$, and expressed by

$$T_m = \frac{a}{\ln(1 - 2f_0) + b}.$$

The slope of the sigmoid, $S = f'(T_i)$, is calculated at the inflection point, $T_i$, where $f''(T_i) = 0$.

The sigmoid function can be reparameterized to fit for the melting temperature directly, by replacing $b$ with $T_m$,

$$f(T) = \frac{1 - f_0}{1 + (1 - 2f_0)^{-1} e^{a\left(\frac{1}{T_m} - \frac{1}{T}\right)}} + f_0.$$

This form of the sigmoid curve was fit to all protein profiles using the nonlinear least squares method (package "nls" in R). Each protein was fit ten times with randomized initial parameters to minimize the chance of finding a false local $\chi^2$ minimum.

To exclude poorly fitting data, Z scores were calculated for the residual mean variation from the curve fitting and included only if |Z score| < 3. Finally, we included only proteins where the standard error of $\Delta T_m$ was less that median + 3 × median average deviation (MAD) for the standard error of $\Delta T_m$. The final list consisted of 3,707 proteins.

### Mass spectrometry analysis of immunoprecipitated proteasome

Protein extracts from each biological replicates were loaded on NuPAGE 4–12% bis–tris acrylamide gels (Invitrogen). Running of protein was stopped as soon as proteins stacked in a single band, stained with InstantBlue (Expedeon), cut from the gel and digested with trypsin protease, MS Grade (Pierce) as described previously (Shevchenko *et al*, 1996). Mass spectrometry analysis was carried out by LC–MS/MS using similar settings as described above except that samples were measured in "Top Speed" data-dependent acquisition mode. Statistical significance was analyzed from the three replicate samples by two-tailed *t*-test without adjusting for multiple hypothesis testing.

### Tandem ubiquitin-binding entity (TUBE) pulldown of ubiquitylated proteins

MCF7 cells were cultured in three replicates with and without 1 μM palbociclib for 24 h. Cells were lysed in 50 mM Tris–HCl pH 7.5, 150 mM NaCl, 1% Triton, 0.1 mM EDTA, 0.1 mM EGTA, 100 mM N-ethylmaleimide and complete protease inhibitor cocktail (Roche). Lysates were sonicated and centrifuged to remove insoluble materials. Protein concentration was determined by BCA protein assay (Pierce), and 10 mg of protein was pre-incubated with 50 μl of HaloLink resin (Promega) for 30 min at 4°C. Supernatants were then incubated with 50 μl of Halolink-UBA resin (UBA domain of Ubiquilin-1, amino acids Q539-Q587) for 3 h at 4°C.

The poly-ubiquitin (polyUb) chains captured by the UBA domain of Ubiquilin (Hjerpe *et al*, 2009) were washed three times with lysis buffer, and Halo-tagged UBA containing polyUb proteins were solubilized by adding reducing Laemmli sample buffer (Invitrogen). Extracts were loaded on NuPAGE 4–12% bis–tris acrylamide. Running of protein was stopped as soon as proteins stacked in a single band, stained with InstantBlue (Expedeon), cut from the gel, reduced with 10 mM TCEP for 10 min at 65°C, and digested with trypsin before analysis by mass spectrometry.

### Preparation of Ub-AQUA peptide mixtures

Concentrated stocks of isotopically labeled internal standard (heavy) peptides and light synthetic peptides (M1, K6, K11, K27, K29, K33, K48, K63) were purchased from Cell Signaling Technologies. All stock solutions were stored at −80°C, working stock solution of individual peptides was prepared at 25 pmol/μl in 2% ACN, 0.1% FA and used to prepare an experimental mixture consisting of eight peptides at 1 pmol/μl in 2% ACN, 0.1% FA. The mixtures were frozen at −80°C in triplicate use aliquots for direct addition to samples.

### Targeted mass spectrometry of ubiquitin chain types

Absolute analysis of ubiquitin chain types was performed similar to reported previously (Huguenin-Dezot *et al*, 2016). Quantitation using parallel reaction monitoring (PRM) was performed on an Orbitrap Fusion mass spectrometer (Thermo Fisher Scientific) with an Easy-Spray source coupled to an Ultimate 3000 Rapid Separation LC system (Thermo Fischer Scientific). Samples were loaded via a 5 μl full loop injection directly onto an EASY-Spray column (15 cm ×

75 μm ID, PepMap C18, 3 μm particles, 100 Å pore size, Thermo Fisher Scientific) and separated by reverse phase chromatography at a flow rate of 1 μl/min where solvent A was 98% $H_2O$, 2% ACN, 0.1% FA and solvent B was 98% ACN, 2% $H_2O$, 0.1% FA. Upon LC direct injection, peptides were resolved with an isocratic gradient of 0.1% of solvent B over 10 min, followed by a step from 0.1 to 25.5% of solvent B over 41 min, 5 min of high organic wash (90% solvent B) and 12 min re-equilibration at 0.1% of solvent B. The Orbitrap Fusion mass spectrometer was operated in targeted mode "tMS2" for the detection of light and synthetic heavy peptides. The included m/z values were selected by the quadrupole, with 4-m/z isolation window, a maximum injection time of 100 ms and a maximum AGC target of 50,000. HCD fragmentation was performed at 30% collision energy for all included peptides, and MS/MS fragments were detected in the Orbitrap mass analyzer at a FMWH resolution of 30,000 (at m/z 200). Peak integration of MS/MS spectra and quantification of Ub peptides were performed on Skyline (version 3.5.0.9191; http://proteome.gs.washington.edu/software/skyline; MacLean *et al*, 2010).

### Kinase inhibitor data

The *in vitro* activity data for palbociclib were downloaded from http://www.kinase-screen.mrc.ac.uk/kinase-inhibitors.

### Kaplan–Meier survival analysis

Kaplan–Meier curves for ECM29 in breast cancer were generated using the KMplot online service (http://www.kmplot.com), which combines publicly available mRNA expression datasets and associated clinical information (Gyorffy *et al*, 2010). The cohorts are divided into high/low according to the median expression of the gene. The survival was assessed for the whole population, and various subpopulations of which HER2 expressing and endocrine therapy receiving patient populations displayed most significant survival benefits.

### Data availability

Mass spectrometric raw data and the MaxQuant outputs are available through the PRIDE repository (https://www.ebi.ac.uk/pride/archive/) and have been assigned the identifier PXD003704.

**Expanded View** for this article is available online.

### Acknowledgements

We would like to thank Philipp Kaldis for comments; DNA cloning, Protein Production, Antibody Production, DNA sequencing facility, and mass spectrometry teams of the MRC Protein Phosphorylation and Ubiquitylation Unit for their support; Victoria Cowling, Gopal Sapkota, Tom Macartney, and Dario Alessi for providing reagents and Orsolya Bilkei-Gorzo for preparing the UBA binding domain of ubiquilin1. This work was funded by Medical Research Council UK (MC_UU_12016/5 to M.T.), the BBSRC (BB/L008874/1), and the pharmaceutical companies supporting the Division of Signal Transduction Therapy (DSTT) (Astra-Zeneca, Boehringer-Ingelheim, GlaxoSmithKline, Janssen Pharmaceuticals, Merck KGaA, and Pfizer). M.T. thanks Thermo-Fisher Scientific for support with the TMT Research Award. The School of Life Sciences Data Analysis Group is funded by the Wellcome Trust grant 097945/Z/11/Z. T.P.M. is supported by the Wellcome Trust Sir Henry Postdoctoral Fellowship (grant number 110275/Z/15/Z). V.M.J is supported by the Conquer Cancer Foundation of ASCO Young Investigator Award (8364), Komen Post-Doctoral Award (15329319), and the Vanderbilt Clinical Oncology Research Career Development Program (2K12CA090625-17).

### Author contributions

Conceptualization: MB and MT together with TPM and JP; Methodology: TPM, JP, AH, MG, VMJ, MB, and MT; Investigation: TPM, JP, AH, and MB; Writing—original Draft: MB; Writing—review & editing: TPM, JP, AH, VMJ, MB, and MT; Supervision: MB and MT.

### Conflict of interest

The authors declare that they have no conflict of interest.

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
