## [Review Process File · The EMBO Journal]

Thermal proteome profiling of breast cancer cells reveals proteasomal activation by CDK4/6 inhibitor palbociclib

Teemu P. Miettinen, Julien Peltier, Anetta Härtlova, Marek Gierliński, Valerie M. Jansen, Matthias Trost and Mikael Björklund

Review timeline:

Submission date:	9 th October 2017
Editorial Decision:	30 th October 2017
Revision received:	6 th February 2018
Editorial Decision:	19 th February 2018
Revision received:	2 nd March 2018
Accepted:	9 th March 2018

Editor:

Transaction Report:

1st Editorial Decision

30th October 2017

Thank you for the submission of your manuscript (EMBOJ-2017-98359) to The EMBO Journal. Your study has been sent to three referees, and we have received reports from all of them, which I copy below.

As you will see, the referees acknowledge the potential high interest and novelty of your work, although they also express a number of concerns that will have to be addressed before they can support publication of your manuscript in The EMBO Journal. In particular, referee #2 points out the need for you to corroborate the senescence phenotype observed and consolidate your findings on implication of ECM29 by gain-of-function analyses. Referee #1 is concerned about the CD4/6-dependence of palbociclib's effects and asks you to investigate further this aspect by additional experiments. In addition, all referees list a number of technical issues on assays used and controls made, that need to be addressed to achieve the level of robustness needed for The EMBO Journal.

I judge the comments of the referees to be generally reasonable and we are in principle happy to invite you to revise your manuscript experimentally to address the referees' comments.

REFeree REPORTS

Referee #1:

The manuscript by Miettinen et al applies the recently developed Thermal Proteome Profiling (TPP) technology to elucidate the mode of action of the CDK inhibitor palbociclib. The authors find that cognate targets of palbociclib are affected in their thermal stability; additionally they observe stabilization of a number of other kinases. Intriguingly, they also find that the 20S proteasome complex changes its stability. The finding is followed and the following points are convincingly demonstrated:

- 1) Palbociclib treatment leads to activation the proteasome
- 2) The activation is not a consequence of inhibition of the cognate CDK targets.

- 3) The activation is due to dissociation of the proteasome with ECM29
- 4) ECM29 RNA levels turn out to some extent to be predictive of relapse free survival of endocrine treated breast cancer patients.

I am convinced by the biological findings and validations, and I find the fact that TPP can discern proteasome activity an exciting development of the technology. It still remains to be understood exactly what is the molecular mechanism of proteasome activation by palbociclib, but in my opinion it is too much to ask for this particular manuscript.

I recommend publication following some minor, but important changes to the manuscript:

From the manuscript: "When combined with mass spectrometry, the thermal stability changes in proteins can be quantified using a quantitative proteomic approach, holding great promise for 110 identifying drug targets in live cells (Franken, Mathieson et al., 2015, Savitski et al., 2014)."

Please be more complete in citations and also cite Becher et al Nat Chem Bio 2016 as this is a recent example of successful in situ off target identification and validation.

From the manuscript:

"Heat aggregated proteins were 135 removed by centrifugation and soluble proteins were analyzed by multiplexed quantitative mass spectrometry using isotopically labelled tandem mass tags."
Please add relevant references for TMT labeling

From the manuscript:

"When equal amount of protein is analyzed by mass spectrometry for each temperature point, thermally stable proteins display increased abundance with temperature in the raw data (Fig. 1a, middle)."

This is an interesting approach for normalization. One point I have is that at 65C you have very little protein left in the soluble fraction. If you want to take the same protein amount there as for 37C, you need much more starting material (number of cells), could you please comment on that?

From the manuscript:

"The remaining proteins with poorer quality thermal denaturation curves were enriched for membrane proteins and mitochondrial proteins as shown before (Savitski et al., 2014)"

This issue has since been resolved by using the mild detergent NP40 instead of only PBS for cell lysate. This has no impact on the conclusion drawn in this study. But I would still ask the authors to clarify that. The way the above sentence reads suggests that currently it is not possible to properly assess membrane proteins, which is not true.

Referee #2:

The paper (EMBOJ-2017-98359) entitled "Thermal proteome profiling of breast cancer cells reveals proteasomal activation by CDK4/6 inhibitor palbociclib" uses thermal profiling as a screening technique to better understand the molecular mechanism of the chemotherapeutic drug palbociclib. This is an important question because a more complete characterization of drug action can be used to identify patients most likely to benefit from its use. Overall this is an impressive paper. The amount of experimental data is extensive and of high quality, with well controlled experiments presented in a logical fashion. The results support the contention that a major unanticipated effect of palbociclib is proteasome activation, and further provides a mechanistic explanation for this phenomenon. A testament to its persuasion is that one is left wondering whether palbociclib designation as a CDK4/6 inhibitor is at all relevant to its clinical utility. The conclusions are significant because they can immediately impact which patient cohorts receive palbociclib, with the potential to enhance its efficacy and extend patient lifespan. In my opinion the manuscript is immediately acceptable for publication with no major changes. Minor comments and interesting questions (which do not need to be answered for publication) are listed below.

I am not by any means an expert on thermal profiling, and the following probably reflects my lack of understanding. However, it seems remarkable that this approach worked, given that it evaluates ligand induced protein thermal stability (105-107). While this can be indirect, in this case the effect (unidentified) results in dissociation of ECM29, which in turn results in thermal shifting of the 20S (not 19S) proteasome subunit (197-199). One can't argue too much with results, but it might be worthwhile to hypothesize why the 20S thermal stability should be affected.

264-266: The lack of increase in protein ubiquitination is curious, given that it is (generally, but not always) a pre-requisite to degradation. So, how can protein degradation increase if ubiquitinated proteins are not being generated at a higher rate? This scenario implies that proteolysis per se is rate limiting, and further that ubiquitinated proteins are lying around waiting to be degraded. Most work in the field suggests otherwise.

There are two general ways to interpret increased proteasome activity against artificial substrates. 1) More proteasomes are being inactivated from an inactive state (such that the total number of active proteasomes increases; or 2) Enhancing the catalytic rate of individual proteasomes (the number of total active proteasomes does not increase). Which explanation do the authors prefer?

400: Is ECM29 upregulated or showing enhanced association with the proteasome in the resistant cell lines?

Given the multiple effects of palbociclib and questions about the significance of CDK4/6 inhibition, it might be worthwhile to determine if palbociclib still has the observed effect on the proteasome in the absence of CDK4/6 (fig 4). In other words, perhaps the effect of palbociclib on CDK4/6 is not what we think it is (in cells).

Minor corrections/comments

31: a phenotype that are is not....

351: lower expression of ECM29 was associated

402: and, at least in MCF7, cells,.....

440: Nonetheless, the palbociclib-induced effect? Augmented proteolysis...

458: In contrast to recent work...This is not really a contrast, and the respective observations are compatible given the complex role of the proteasome in cell cycle progression.

Referee #3:

1-

General summary and opinion about the principle significance of the study, its questions and findings

In this manuscript, Miettinen et al. use a thermal proteome profiling approach to identify molecular mechanism(s) responsible for palbociclib-induced senescence in ER-positive breast cancer cells. Interestingly authors found that CDK4/6 inhibition activates the proteasome by an indirect mechanism that is mediated by the decreased abundance of the proteasomal inhibitor ECM29, ultimately inducing senescence.

The study is conceptually relevant, given the important that CDK4/6 currently have in the clinic and the need of better understanding the anti-cancer mechanisms these drugs. In general the conclusions are supported by the results. However the work raise some questions that should be addressed prior its publication in The EMBO Journal.

2-

specific major concerns essential to be addressed to support the conclusions

Despite being an interesting discovery, particularly given its clinical relevance, the implication of ECM29 in the senescence induction lack some depth. The reduced proliferation and enlarged cell morphology observed upon ECM29 loss indeed suggests features of senescence (Line 337 & Fig. 5d). However, when used as a single method, acidic beta-galactosidase activity might not be a reliable senescent marker, as it may simply indicate general cellular stress. In order to draw solid conclusions, authors should confirm the senescent state secondary to ECM29 loss with additional established senescence molecular markers (e.g., reduced RB phosphorylation, induction of p16, p21, Lamin B1 suppression, upregulation of SASP members).

The suggestion that patients with high levels of ECM29 would benefit the most for palbociclib treatment should be strengthened with gain of function experiments and possibly validated in vivo.

The methods section lacks detail on how the CRISPR edited MCF7 ECM29^{+/-} cells were generated, if by stable integration or by transient transfection of the vectors into the cells. The appropriate control (particularly in the case of a stable integration) in Fig. 5c would be an MCF7 cell line edited with CRISPR/Cas9 for a non-targeted sequence delivered in the exact same way as for the ECM29^{+/-} cells.

Authors did confirm the reduced proliferation phenotype of the ECM29 loss with siRNA using the appropriate siRNA control in Supplementary Fig. 7b. However, the level of ECM29 knockdown in these experiments is missing.

Given that the authors were not able to generate ECM29 homozygous knockout cells (most likely due to the lethal effects), did authors verified that the observed reduced proliferation illustrated in Fig. 5c and Supplementary Fig. 7b is not due to cell death/apoptosis of the gRNA-ECM29 or siRNA-ECM29 cells?

What was the cut off used to define high versus low ECM29 mRNA expression (Fig. 5e-g)?

What was the methodology used in the cell cycle analysis illustrated in Fig. 6c-e?

3-
minor concerns that should be addressed
Line 31 - "is" instead of "are"

Line 331 - Although the levels of ECM29 are about 50% reduced comparing to the WT cells (Western blot in Fig. 5c), it is not clear to the reader how did the authors conclude that the MCF7 CRISPR edited cells are heterozygous for ECM29 without a proper sequencing experiment to prove that.

Line 1055 - "levels from Western blots in (g)" instead of "(e)"

Response to reviewers regarding the manuscript by Miettinen, Peltier, et al.

The comments made by reviewers are in black and our responses are in red. Thank you for taking the time to review our work and provide constructive feedback. In the manuscript all changes have been highlighted with yellow colour.

Referee #1:

The manuscript by Miettinen et al applies the recently developed Thermal Proteome Profiling (TPP) technology to elucidate the mode of action of the CDK inhibitor palbociclib. The authors find that cognate targets of palbociclib are affected in their thermal stability, additionally they observe stabilization of a number of other kinases. Intriguingly, they also find that the 20S proteasome complex changes its stability. The finding is followed and the following points are convincingly demonstrated:

- 1) Palbociclib treatment leads to activation the proteasome
- 2) The activation is not a consequence of inhibition of the cognate CDK targets.
- 3) The activation is due to dissociation of the proteasome with ECM29
- 4) ECM29 RNA levels turn out to some extent to be predictive of relapse free survival of endocrine treated breast cancer patients.

I am convinced by the biological findings and validations, and I find the fact that TPP can discern proteasome activity an exciting development of the technology. It still remains to be understood exactly what is the molecular mechanism of proteasome activation by palbociclib, but in my opinion it is too much to ask for this particular manuscript.

We are grateful for the positive comments and, as shown below, we have addressed all the points raised by the reviewer.

I recommend publication following some minor, but important changes to the manuscript:

From the manuscript:

"When combined with mass spectrometry, the thermal stability changes in proteins can be quantified using a quantitative proteomic approach, holding great promise for identifying drug targets in live cells (Franken, Mathieson et al., 2015, Savitski et al., 2014)."

Please be more complete in citations and also cite Becher et al Nat Chem Bio 2016 as this is a recent example of successful in situ off target identification and validation.

We have now cited this paper as suggested.

From the manuscript:

"Heat aggregated proteins were removed by centrifugation and soluble proteins were analyzed by multiplexed quantitative mass spectrometry using isotopically labelled tandem mass tags."

Please add relevant references for TMT labelling

We now refer in this context the original paper by (Savitski et al., 2014), where the same TMT labelling approach was used as requested.

From the manuscript:

"When equal amount of protein is analyzed by mass spectrometry for each temperature point, thermally stable proteins display increased abundance with temperature in the raw data (Fig. 1a, middle)."

This is an interesting approach for normalization. One point I have is that at 65C you have very little protein left in the soluble fraction. If you want to take the same protein amount there as for 37C, you need much more starting material (number of cells), could you please comment on that?

We agree that at the highest temperatures, relatively little protein is left soluble. While this could potentially cause sensitivity issues for the mass spec analysis, it should have minor effect on the data normalization and fitting. We wanted to keep it simple and consistent by analysing same amount of starting material.

From the manuscript:

"The remaining proteins with poorer quality thermal denaturation curves were enriched for membrane proteins and mitochondrial proteins as shown before (Savitski et al., 2014)"

This issue has since been resolved by using the mild detergent NP40 instead of only PBS for cell lysate. This has no impact on the conclusion drawn in this study. But I would still ask the authors to clarify that. The way the above sentence reads suggest that currently it is not possible to properly assess membrane proteins, which is not true.

The reviewer is correct and we have now acknowledged this in the introduction by stating that "Furthermore, alternative sample preparation with NP40 has allowed improved detection of membrane proteins (Reinhard et al., 2015)."

Referee #2:

The paper (EMBOJ-2017-98359) entitled "Thermal proteome profiling of breast cancer cells reveals proteasomal activation by CDK4/6 inhibitor palbociclib" uses thermal profiling as a screening technique to better understand the molecular mechanism of the chemotherapeutic drug palbociclib. This is an important question because a more complete characterization of drug action can be used to identify patients most likely to benefit from its use. Overall this is an impressive paper. The amount of experimental data is extensive and of high quality, with well controlled experiments presented in a logical fashion. The results support the contention that a major unanticipated effect of palbociclib is proteasome activation, and further provides a mechanistic explanation for this phenomenon. A testament to its persuasion is that one is left wondering whether palbociclib designation as a CDK4/6 inhibitor is at all relevant to its clinical utility. The conclusions are significant because they can immediately impact which patient cohorts receive palbociclib, with the potential to enhance its efficacy and extend patient lifespan. In my opinion the manuscript is immediately acceptable for publication with no major changes. Minor comments and interesting questions (which do not need to be answered for publication) are listed below.

We thank the reviewer for the very positive comments. Below we have tried to address all the minor comments raised by the reviewer.

I am not by any means an expert on thermal profiling, and the following probably reflects my lack of understanding. However, it seems remarkable that this approach worked, given that it evaluates ligand induced protein thermal stability (105-107). While this can be indirect, in this case the effect (unidentified) results in dissociation of ECM29, which in turn results in thermal shifting of the 20S (not 19S) proteasome subunit (197-199). One can't argue too much with results, but it might be worthwhile to hypothesize why the 20S thermal stability should be affected.

In the absence of structural information, we can only speculate how ECM29 affects the thermal stability of the 20S subunit. Nevertheless, this finding is consistent with prior work showing that ECM29 interaction involves change in proteasome conformation (De La Mota-Peynado, JBC, 2013). We have now added this second point to the manuscript by stating that “ECM29 binding to the 20S proteasome also changes the proteasome conformation (De La Mota-Peynado, Lee et al., 2013), which likely explains the increased 20S thermal stability upon ECM29 dissociation from the proteasome.” A potential way to address this would be through cryo-electron microscopy, but this would be a rather time consuming and laborious follow-up paper by itself based on discussion with an expert in the field.

264-266: The lack of increase in protein ubiquitination is curious, given that it is (generally, but not always) a pre-requisite to degradation. So, how can protein degradation increase if ubiquitinated proteins are not being generated at a higher rate? This scenario implies that proteolysis per se is rate limiting, and further that ubiquitinated proteins are lying around waiting to be degraded. Most work in the field suggests otherwise.

The reviewer is correct that this is an unexpected result. We do not want to claim that for all proteins the rate-limiting step in degradation is proteasome activity. However, substrate unfolding and entry into proteasome particle are also important for proteolysis (Dorn IT, et al., J Mol Biol, 1999. & Henderson A, et al., JBC, 2011). Considering that the presence of EMC29 on the proteasome is believed to close the substrate entry channel of the proteasome (De La Mota-Peynado, Lee et al., 2013), this data suggests that the entry into proteasome can limit the degradation of at least some proteins in a manner that is not inconsistent with the current understanding of ubiquitination being generally rate-limiting. Note that in order to explain our data, not all proteins have to be degraded faster when proteasome is activated. As palbociclib also differentially affects ubiquitin linkages, it is possible that this affects substrate degradation as, for example, K48/K63 branched linkages preferentially associate with proteasomes in cells (Ohtake F, et al., PNAS, 2018).

As this is an important point, we have now tried to clarify our manuscript by adding the following comments to the manuscript:

“As the rate of degradation for most proteins is regulated by ubiquitylation, not proteasome activity, the palbociclib induced proteasomal activation may only affect a specific subset of proteins, such as those with K48/K63 branched linkages (Ohtake, Tsuchiya et al., 2018).”

“Furthermore, our observation that palbociclib increases proteasome activity, but not ubiquitination levels, is consistent with the previous work suggesting that the presence of ECM29 on the proteasome induces a closed conformation of the substrate entry channel of the core particle (De La Mota-Peynado et al., 2013).”

There are two general ways to interpret increased proteasome activity against artificial substrates. 1) More proteasomes are being inactivated from an inactive state (such that the total number of active proteasomes increases; or 2) Enhancing the catalytic rate of individual proteasomes (the number of total active proteasomes does not increase). Which explanation do the authors prefer?

Our data does not directly provide any evidence to separate these two options and both of them might be true. The current literature on ECM29 suggests that at least option #2, where catalytic rate is increased, is likely true (ECM29 inhibits proteasomal degradation rate). At the same time, it may be that not all proteasomes are inhibited by ECM29 under normal conditions. The stoichiometry between ECM29 and proteasomes is not known, nor do we know if there are separate pools of

proteasome with and without ECM29. Thus, palbociclib or ECM29 knockdown may also change the relative abundances of proteasome pools with and without ECM29. While we agree that this is a very thought provoking question, it goes way beyond the scope of this manuscript.

400: Is ECM29 upregulated or showing enhanced association with the proteasome in the resistant cell lines?

Unfortunately, we do not have RNA-seq or MS data on these cells to fully answer this question. However, the resistant cell lines display overactive AKT pathway (Jansen, et al., 2017), which may explain the lower baseline proteasome activity through its effect on mTOR. If this is the case, the mechanism affecting the proteasome is distinct from palbociclib, as our data clearly shows that mTOR inhibition does not induce similar effects on ECM29 as palbociclib. Thus, we think it is unlikely that ECM29 is radically affected in the resistant cells, but rather the resistance is due to the altered baseline proteasomal (or ubiquitin pathway) activity.

Given the multiple effects of palbociclib and questions about the significance of CDK4/6 inhibition, it might be worthwhile to determine if palbociclib still has the observed effect on the proteasome in the absence of CDK4/6 (fig 4). In other words, perhaps the effect of palbociclib on CDK4/6 is not what we think it is (in cells).

This is a good point, which we have considered. As shown in figure 4, we have done this using knockdown experiments, which are not the perfect validation as some CDK activity might remain. However, knockout of both CDK4 and CDK6 completely blocks proliferation in breast cancer cells. This is the very rationale for the development of these Cdk4/6 inhibitors! This makes the use of CRISPR-Cas9 system a non-viable option, as the resulting population cannot be expanded sufficiently to carry out the experiments. We have also considered obtaining CDK4 and CDK6 null MEFs but, unfortunately, the one laboratory currently in the possession of those MEFs has not replied to our emails. While genetic models of Cdk4/6 inhibition clearly indicate that breast cancer cell proliferation is dependent on these Cdk, I think the big issue with palbociclib is that it is not such a specific inhibitor as previous literature made everyone to believe. Palbociclib's effects on proteasome and other kinases (as also demonstrated by a recent Science paper (Klaeger S, The target landscape of clinical kinase drugs. December 2017) showing that palbociclib was one of the most promiscuous kinase inhibitors) clearly demonstrates this. It is likely because of the lack of CDK4/6 specificity that it works so well in the clinic

Minor corrections/comments

31: a phenotype that are is not....

351: lower expression of ECM29 was associated

402: and, at least in MCF7, cells,.....

440: Nonetheless, the palbociclib-induced effect? augmented proteolysis...

458: In contrast to recent work...This is not really a contrast, and the respective observations are compatible given the complex role of the proteasome in cell cycle progression.

Thank you for pointing these minor corrections out to us.

Referee #3:

1-

general summary and opinion about the principle significance of the study, its questions and

findings

In this manuscript, Miettinen et al. use a thermal proteome profiling approach to identify molecular mechanism(s) responsible for palbociclib-induced senescence in ER-positive breast cancer cells. Interestingly authors found that CDK4/6 inhibition activates the proteasome by an indirect mechanism that is mediated by the decreased abundance of the proteasomal inhibitor ECM29, ultimately inducing senescence.

The study is conceptually relevant, given the important that CDK4/6 currently have in the clinic and the need of better understanding the anti-cancer mechanisms these drugs. In general the conclusions are supported by the results. However the work raise some questions that should be addressed prior its publication in The EMBO Journal.

We would like to thank the reviewer for positively acknowledging that our work is relevant and conclusion are supported by the data. Below we have addressed the points raised by the reviewer.

2-

specific major concerns essential to be addressed to support the conclusions

Despite being an interesting discovery, particularly given its clinical relevance, the implication of ECM29 in the senescence induction lack some depth. The reduced proliferation and enlarged cell morphology observed upon ECM29 loss indeed suggests features of senescence (Line 337 & Fig. 5d). However, when used as a single method, acidic beta-galactosidase activity might not be a reliable senescent marker, as it may simply indicate general cellular stress. In order to draw solid conclusions, authors should confirm the senescent state secondary to ECM29 loss with additional established senescence molecular markers (e.g., reduced RB phosphorylation, induction of p16, p21, Lamin B1 suppression, upregulation of SASP members).

We agree with the reviewer and to better address the role of ECM29 in senescence we have repeated our ECM29 knockdowns in MCF7 and performed additional senescence marker assays. In short, we examined two independent senescence markers, p21 protein levels and S139 phosphorylation levels of H2AX. Both of these markers were significantly upregulated as a consequence of ECM29 knockdown. These data have been added to Figure 5 (new panels e-g) and the following text has been added to the manuscript: “We further validated this by examining two other senescence associated markers, S139 phosphorylation of Histone H2A.X (γ H2AX) and p21 protein levels (Lawless, Wang et al., 2010), in MCF7 cells where ECM29 was silenced using with siRNAs (Fig. 5e). Both of these markers displayed marked increase after silencing of EMC29 (Fig. 5f and Fig 5g) supporting our other results.”

The suggestion that patients with high levels of ECM29 would benefit the most for palbociclib treatment should be strengthened with gain of function experiments and possibly validated *in vivo*.

Again, we agree with the reviewer that gain-of-function experiments would strengthen our conclusions and we have carefully considered such experiment. However, we have not been able to obtain a gain-of-function model. This is likely due to two facts. First, ECM29 is a huge protein, approximately 205 kDa. This makes transfections and controlling the expression level of the protein ineffective. Second, even if proper expression levels were achieved, the ECM29 association with the proteasome may not increase which would be the pre-requisite for a gain-of-function phenotype. The mechanism which controls ECM29 association with the proteasome is not known and we thus do not have the required knowledge to create a gain-of-function system. We further believe that - as much as we would like these experiments - *in vivo* data would be out of the scope of this paper.

The methods section lacks detail on how the CRISPR edited MCF7 ECM29^{+/-} cells were generated, if by stable integration or by transient transfection of the vectors into the cells. The appropriate control (particularly in the case of a stable integration) in Fig. 5c would be an MCF7 cell line edited with CRISPR/Cas9 for a non-targeted sequence delivered in the exact same way as for the ECM29^{+/-} cells.

Thank you for pointing this out. We generated these cell lines using transient transfection and thus we believe that the WT cells without transfection of Cas9 and a non-targeted sequence represents a more accurate control. We have now added further details of the CRISPR cells in the methods section. We generated two independent cell lines both displaying approximately 50% reduction and show data with only one of these as they performed identically in preliminary tests.

Authors did confirm the reduced proliferation phenotype of the ECM29 loss with siRNA using the appropriate siRNA control in Supplementary Fig. 7b. However, the level of ECM29 knockdown in these experiments is missing.

We have now added western blots displaying the knockdown efficiency after 48h RNAi. These siRNA mediated knockdown worked robustly in our hands and we did not therefore see the need to quantify the knockdown levels each day of the experiment.

Given that the authors were not able to generate ECM29 homozygous knockout cells (most likely due to the lethal effects), did authors verified that the observed reduced proliferation illustrated in Fig. 5c and Supplementary Fig. 7b is not due to cell death/apoptosis of the gRNA-ECM29 or siRNA-ECM29 cells?

The RNAi, knock-out and drug experiments indicate that the cells do not proliferate as they exit the cell cycle and become quiescent rather than die. When analysing gRNA-ECM29 or siRNA-ECM29 cells, we did not observe any significant cell death/apoptosis resembling changes as measured by flow cytometry. Our new western blots for p21 and S139 phosphorylation levels of H2AX further support this.

What was the cut off used to define high versus low ECM29 mRNA expression (Fig. 5e-g)?

We used the default settings for the kmplot software. The cohorts are divided into two high/low according to the median expression of the measured mRNA.

What was the methodology used in the cell cycle analysis illustrated in Fig. 6c-e?

The method used was EtOH fixation followed by RNase treatment and PI staining. We have now clarified this in our methods section.

3-
minor concerns that should be addressed
Line 31 - "is" instead of "are"

Thank you for pointing this out, the mistake has now been corrected.

Line 331 - Although the levels of ECM29 are about 50% reduced comparing to the WT cells (Western blot in Fig. 5c), it is not clear to the reader how did the authors conclude that the MCF7 CRISPR edited cells are heterozygous for ECM29 without a proper sequencing experiment to prove that.

The reviewer is absolutely correct in pointing this out. Unfortunately, when these experiments were done we did not have the resources to carry out the sequencing validation of the heterozygous state at the gene level, and the state is assumed based on protein expression levels shown in western blots. We have now corrected the text to be more precise about this. However, we would like to highlight that our conclusion that ECM29 is required for maintaining proliferation in breast cancer cells remains valid, as this experiment together with the siRNA based experiments show a substantial reduction in proliferation after targeting ECM29.

Line 1055 - "levels from Western blots in (g)" instead of "(e)"

Thank you for pointing this out, the mistake has now been corrected.

Thank you for submitting your revised manuscript for consideration by The EMBO Journal. It has now been seen by the three original referees, whose comments are enclosed below. As you will see, all referees find that their concerns have been sufficiently addressed and are now broadly in favour of publication, pending minor issues brought up by referee #3 are convincingly addressed.

Thus, we are pleased to inform you that your manuscript has been accepted in principle for publication in The EMBO Journal, pending satisfactory revision of the remaining issues related to data representation.

REFeree REPORTS

Referee #1:

I am very satisfied with how the authors addressed my comments and support publication.

Referee #2:

My previous review of this paper was highly positive (a relative rarity), and that opinion has not changed. I have evaluated the authors' response to other reviewers, and think they have responded appropriately to address their concerns. In my view the manuscript is suitable for publication.

Referee #3:

Miettinen et al. provide a revised manuscript in which the great majority of the comments and questions were addressed convincingly. However, the demonstration that the loss of ECM29 induces senescence remains insufficiently proven. Currently, there is no gold standard method for senescence detection and thus the molecular characterization of cellular senescence relies on the combined analysis of a panel of markers (see for example PMID: 27979832). In this revised version authors claim senescence induction based on the analysis of beta-galactosidase staining and a modest increase of phosphorylation of histone H2AX and p21 protein levels. Given the use of only three markers and the technical limitations of the experimental design (i.e., cell extracts harvested only 48h after ECM29 silencing, instead of the usual 5-7 days of senescence-inducing treatments), authors should tone down the claim of cellular senescence induction upon ECM29 loss. Instead, and in accordance with the provided data, ECM29 loss seems to induce a proliferation arrest phenotype associated with few senescence markers.

The beta-galactosidase staining could be illustrated with better quality pictures, e.g. WT control without the intense blue background. In addition, the legend of the western blot quantification in panels (f) and (g) should be more clear in explaining the standard deviation of the n=3. Are these 3 independent experiments? Three technical replicates? Three quantifications of the same band? Finally, it is not clear which loading control was used in this western blot experiment: beta-tubulin (indicated in the western blot) or alpha-tubulin (indicated in the quantification)?

We have performed all the requests of reviewer 3 and toned down the senescence induction throughout the abstract and the manuscript and added information about the number of independent experiments.

Corresponding Author Name: Mikael Bjorklund & Matthias Trost

Manuscript Number: EMBOJ-2017-98359